# Smoothed Normalization for Efficient Distributed Private Optimization

## Abstract

Federated learning enables training machine learning models while preserving the privacy of participants. Surprisingly, and to the best of our knowledge, there is no differentially private distributed method for smooth non-convex optimization problems. The reason is that standard privacy techniques require bounding the participants' contributions, usually enforced via *clipping* of the updates. Existing literature typically ignores the effect of clipping by assuming the boundedness of gradient norms or analyzes distributed algorithms with clipping but ignores DP constraints. In this work, we study an alternative approach via *smoothed normalization* of the updates motivated by its favorable performance in the centralized setting. By integrating smoothed normalization with an error-feedback mechanism, we design a new distributed algorithm $\alpha$-NormEC. We prove that our method achieves a superior convergence rate over prior works. By extending $\alpha$-NormEC to the DP setting, we obtain the first differentially private distributed optimization algorithm with provable convergence guarantees. Finally, we support our theoretical findings with experiments on practical machine learning problems.

## 1. Introduction

Federated Learning (FL) (Konečný et al., 2016; McMahan et al., 2017; 2018) has become a viable approach for distributed collaborative training of modern machine learning models (He et al., 2015; Ganesh et al., 2019; Silver et al., 2016). This growing interest has spurred the development of novel distributed optimization methods tailored for FL, focusing on ensuring high *communication efficiency* (Kairouz et al., 2021). Although FL optimization methods ensure that private data is never directly transmitted, Boenisch

et al. (2023) demonstrated that the global models produced through FL can still enable the reconstruction of clients' data individually. Therefore, it is essential to study *differentially private* distributed optimization methods for differentially private training (Dwork et al., 2014; McMahan et al., 2018; Sun et al., 2019).

To address emerging privacy risks in FL, differential privacy (DP) (Dwork et al., 2014) has become the standard for providing theoretical privacy guarantees in optimization methods. To enfore DP, clipping is employed. It bounds gradient sensitivity, allowing the addition of DP noise to the updates before communication. One common DP gradient method with clipping is Differentially Private Stochastic Gradient Descent (DP-SGD) (Abadi et al., 2016). However, even in the non-private setting, DP-SGD can hinder convergence, due to the bias introduced by clipping (Koloskova et al., 2023). Often, distributed DP gradient methods with clipping have been studied in the private setting under assumptions that are unrealistic for heterogeneous FL environments, such as bounded gradients (Li et al., 2022; Wang et al., 2023; Lowy et al., 2023; Zhang et al., 2020), which effectively ignore the impact of clipping bias. To our knowledge, no existing distributed DP gradient method has been shown to converge for non-convex, smooth problems without inadequately handling or disregarding the clipping bias.

Error Feedback (EF) mechanisms, also known as Error Compensation (EC), such as EF21 (Richtárik et al., 2021) have been employed to mitigate the clipping bias and achieve strong convergence in the non-private setting, as studied by Khirirat et al. (2023); Yu et al. (2023). However, extending these methods to the private setting is still an open problem. Furthermore, as the clipping threshold highly affects the convergence speed and the DP noise variance, optimizing the convergence of distributed DP clipping methods requires an extensive grid search to determine the appropriate clipping threshold. This process can be computationally expensive (Andrew et al., 2021), and lead to additional privacy loss (Papernot & Steinke, 2021). To address the need for manually tuning the clipping threshold, two major approaches have emerged. The first approach is to use adaptive clipping techniques, such as adaptive quantile clipping, initially proposed by Andrew et al. (2021) and further analyzed by Merad & Gaïffas (2023); Shulgin & Richtárik (2024). The second approach, which is the focus in this paper, is to

[1]Anonymous Institution, Anonymous City, Anonymous Region, Anonymous Country. Correspondence to: Anonymous Author <anon.email@domain.com>.

Preliminary work. Under review by the International Conference on Machine Learning (ICML). Do not distribute.

replace clipping with normalization-like operator.

**Smoothed normalization** originally introduced by Bu et al. (2024); Yang et al. (2022), serves as an alternative to clipping. Unlike clipping, smoothed normalization eliminates the need for manually tuning the clipping threshold. By ensuring that the Euclidean norm of the normalized gradient is bounded above by one, smoothed normalization guarantees robust performance of DP-SGD in convergence and privacy. However, there is very limited literature that characterizes properties of smoothed normalization, and a rigorous convergence analysis for DP-SGD using this operator especially in the distributed setting. While the method has been studied in the single-node setting by Bu et al. (2024) and Yang et al. (2022), the convergence results rely on unrealistic and/or restrictive assumptions, such as symmetric gradient noise (Bu et al., 2024) and almost sure bounds on the gradient noise variance (Yang et al., 2022).

### 1.1. Contributions

Inspired by the success of error feedback and smoothed normalization, we propose $\alpha$-NormEC. Our method provides, for the first time, convergence guarantees in the DP setting without bounded gradient norm assumptions that are typically imposed in prior work. Our detailed contributions are summarized as follows:

• **Favorable properties of smoothed normalization.** In Section 3.3, we present the novel properties of smoothed normalization. We show that smoothed normalization enjoys a "contractive" property similar to biased compression operators (Beznosikov et al., 2023) widely used for reducing communication in distributed learning. This property essentially allows for designing $\alpha$-NormEC that combines smoothed normalization with error feedback.

• **Convergence for non-convex, smooth problems without bounded gradient norm assumptions.** In Section 4, we prove that $\alpha$-NormEC achieves optimal convergence rate (Carmon et al., 2020) for minimizing non-convex, smooth functions without imposing additional restrictive assumptions, such as bounded gradient norms or bounded heterogeneity. Specifically, hyperparameters for tuning $\alpha$-NormEC are easy to implement, in contrast to the stepsize of Clip21 (Khirirat et al., 2023) that depends on the inaccessible value of $f(x^0) - f^{\inf}$. Furthermore, $\alpha$-NormEC with properly tuned hyperparameters achieves a faster convergence rate than Clip21.

• **The first provable convergence in the private setting under standard assumptions.** In Section 5, we extend $\alpha$-NormEC to the differential privacy (DP) setting. Specifically, $\alpha$-NormEC achieves the first convergence guarantees for DP, non-convex, smooth problems *without* ignoring the bias introduced by clipping/normalization. This is the first

provably efficient distributed method in the DP setting under standard assumptions, thus addressing the theoretical gap left by prior work such as Khirirat et al. (2023); Yu et al. (2023), which did not adapt distributed gradient clipping methods for private training.

• **Robust empirical convergence of $\alpha$-NormEC.** In Section 6, we verify the theoretical benefits of $\alpha$-NormEC in both non-private and private settings via numerical experiments on the image classification task with the CIFAR-10 dataset using the ResNet20 model. We demonstrate that $\alpha$-NormEC achieves robust convergence performance across a wide range of its tuning parameters. Furthermore, $\alpha$-NormEC outperforms distributed methods with direct smoothed normalization in convergence speed and accuracy.

## 2. Related Work

**Clipping and normalization.** In machine learning, clipping and normalization address many key challenges. They mitigate the problem of exploding gradients in recurrent neural networks (Pascanu, 2013), enhance neural network training for tasks in natural language processing (Merity et al., 2017; Brown et al., 2020) and computer vision (Brock et al., 2021), ensure privacy in differentially private machine learning (Abadi et al., 2016; McMahan et al., 2018), and stabilize training in the presence of misbehaving or adversarial workers (Karimireddy et al., 2021; Özfatura et al., 2023; Malinovsky et al., 2023). In this paper, we consider smoothed normalization, recently introduced by Bu et al. (2024); Yang et al. (2022), as an alternative to clipping, offering its hyperparameter that supports robust empirical performance in the DP setting.

**Private optimization methods.** DP-SGD (Abadi et al., 2016) is the common first-order method that achieves the DP guarantee by clipping (or normalizing) the gradient before adding noise scaled with the clipped gradient's sensitivity. However, existing DP-SGD convergence analyses often neglect the clipping bias. Specifically, convergence results for smooth functions under differential privacy often require either the assumption of bounded gradients (Zhang et al., 2020; Li et al., 2022; Zhang et al., 2022; Wang et al., 2023; Lowy et al., 2023; Murata & Suzuki, 2023; Wang et al., 2024) or conditions where clipping is effectively inactive (Zhang et al., 2024; Noble et al., 2022). Thus, in this analytical approach, the convergence behaviors of DP-SGD are not fully understood.

**Single-node non-private methods with clipping.** The impact of clipping on single-node gradient methods for non-private optimization has been extensively studied. Numerous works have shown strong convergence guarantees

of clipped gradient methods under various conditions, including nonsmooth, rapidly growing convex functions Shor (2012); Ermoliev (1988); Alber et al. (1998), generalized smoothness (Zhang et al., 2019; Koloskova et al., 2023; Gorbunov et al., 2024; Vankov et al., 2024; Lobanov et al., 2024; Hübler et al., 2024b), and heavy-tailed noise (Gorbunov et al., 2020a; Nguyen et al., 2023; Gorbunov et al., 2023; Hübler et al., 2024a; Chezhegov et al., 2024).

**Distributed non-private methods with clipping.** Applying gradient clipping in the distributed setting is a challenging task. Existing convergence analyses often rely on bounded heterogeneity assumptions, which often do not hold in cases of arbitrary data heterogeneity. For example, federated optimization methods with clipping have been analyzed under the bounded difference between the local and global gradients (Wei et al., 2020; Liu et al., 2022; Crawshaw et al., 2023; Li et al., 2024). However, even in the non-private setting, these distributed clipping methods do not converge for solving simple problems (Chen et al., 2020; Khirirat et al., 2023). To address the convergence issue, one approach is to use error feedback mechanisms, such as EF21 (Richtárik et al., 2021), as employed by Khirirat et al. (2023); Yu et al. (2023), to compute local gradient estimators and alleviate clipping bias. However, these distributed clipping methods using error feedback are limited to the non-private setting under arbitrary heterogeneity conditions, and extending the methods to the DP setting is still an open problem. In this paper, we propose a distributed method that replaces clipping with smoothed normalization in the EF21 mechanism. Unlike Clip21 (Khirirat et al., 2023), our method provides the first provable convergence guarantees in the DP setting, and empirically outperforms the distributed, deterministic version of DP-SGD with smoothed normalization Bu et al. (2024); Yang et al. (2022), a special case of Das et al. (2021) (with a single local step).

**Error feedback.** Error feedback, or error compensation, has been applied to improve the convergence of distributed methods with gradient compression for communication-efficient learning. First introduced by Seide et al. (2014), EF14 was extensively analyzed for first-order methods in both single-node (Stich et al., 2018; Karimireddy et al., 2019; Stich & Karimireddy, 2019; Khirirat et al., 2019) and distributed settings (Wu et al., 2018; Alistarh et al., 2018; Gorbunov et al., 2020b; Qian et al., 2021; Tang et al., 2019; Danilova & Gorbunov, 2022; Qian et al., 2023). Another error feedback variant is EF21 proposed by Richtárik et al. (2021) that ensures strong convergence under any contractive compression operator for non-convex, smooth problems. Recent variants, e.g. EF21-SGD2M (Fatkhullin et al., 2024) and EControl (Gao et al., 2023) have been developed to obtain the lower iteration and communication complexities than EF21 for stochastic optimization.

## 3. Preliminaries

### 3.1. Notations

We define $[a, b] := \{a, a+1, a+2, \ldots, b\}$ for integers $a, b$ such that $a \leq b$. The expectation of a random variable $u$ is denoted by $\mathrm{E}[u]$. Furthermore, $\langle x, y \rangle$ represents the inner product between $x$ and $y$ in $\mathbb{R}^d$, and the Euclidean norm of $x \in \mathbb{R}^d$ is given by $\|x\| := \sqrt{\langle x, x \rangle}$. Finally, we use the standard order notation $\mathcal{O}(\cdot)$ to hide absolute constants.

### 3.2. Problem Formulation

We focus on solving the finite-sum optimization problem:

$$\min_{x \in \mathbb{R}^d} \left\{ f(x) := \frac{1}{n} \sum_{i=1}^{n} f_i(x) \right\}, \qquad (1)$$

where $x \in \mathbb{R}^d$ is the vector of model parameters of dimension $d$, and $f_i : \mathbb{R}^d \to \mathbb{R}$ is either a loss function on client $i \in [1, n]$ (distributed setting) or data point $i$ (single-node setting). Moreover, we impose the following assumption on objective functions that are standard for analyzing the convergence of first-order optimization algorithms (Nesterov et al., 2018).

**Assumption 1.** *Let the function $f : \mathbb{R}^d \to \mathbb{R}$ be bounded from below by a finite constant $f^{\mathrm{inf}}$, i.e. $f(x) \geq f^{\mathrm{inf}} > -\infty$ for all $x \in \mathbb{R}^d$, and be L-smooth, i.e. $\|\nabla f(x) - \nabla f(y)\| \leq L \|x - y\|$ for all $x, y \in \mathbb{R}^d$.*

*Also, let each component function $f_i : \mathbb{R}^d \to \mathbb{R}$ be $L_i$-smooth, i.e. $\|\nabla f_i(x) - \nabla f_i(y)\| \leq L \|x - y\|$ for all $x, y \in \mathbb{R}^d$.*

### 3.3. DP-SGD

The most common approach to solve Problem (1) under the approximate $(\epsilon, \delta)$-differential privacy (Dwork et al., 2006) is via the DP-SGD method (Abadi et al., 2016)

$$x^{k+1} = x^k - \gamma \left( \frac{1}{B} \sum_{i \in \mathcal{B}^k} \Psi(\nabla f_i(x^k)) + z^k \right), \qquad (2)$$

where $\gamma > 0$ is the stepsize, $\mathcal{B}^k$ is a subset of $\{1, 2, \ldots, n\}$ with cardinality $|\mathcal{B}^k| = B$, $z^k \in \mathbb{R}^d$ is the DP noise, and $\Psi : \mathbb{R}^d \to \mathbb{R}^d$ is an operator with bounded norm, i.e. $\|\Psi(g)\| \leq \Phi$ for any $g \in \mathbb{R}^d$ and some $\Phi > 0$. The method (2) is shown to achieve $(\epsilon, \delta)$-DP by Abadi et al. (2016) if $z^k$ is zero-mean Gaussian noise with variance

$$\sigma_{\mathrm{DP}}^2 \geq \Phi^2 \cdot \frac{cB^2}{n^2} \frac{K \log(1/\delta)}{\epsilon^2}, \qquad (3)$$

where $c > 0$ is a constant, and $K > 0$ is the total number of iterations. A choice to obtain reasonable DP guarantees is to set $\epsilon \leq 10$ and $\delta \ll 1/n$, where $n$ is the number of data

points (Ponomareva et al., 2023). Note that the variance (3) is scaled with the sensitivity $\Phi$.

The method (2) has been often analyzed, e.g. by Zhang et al. (2020; 2022); Murata & Suzuki (2023), under the bounded gradient norm assumption

$$\|\nabla f_i(x)\| \leq \Phi \quad \text{for all } i \text{ and } x \in \mathbb{R}^d. \quad (4)$$

However, this assumption has several limitations. Firstly, it ignores the effect of clipping by setting $\Psi(\cdot)$ as the identity operator. The sensitivity $\Psi$ is usually impossible to compute for many loss functions used in training machine learning models. Even when it can be estimated, its resulting upper bound is often overly pessimistic, leading to excessively large DP noise and thus significantly degrading the algorithmic convergence performance. Secondly, this assumption restricts the class of loss functions $f$. For instance, it does hold for simple quadratic functions over unbounded domain. Thirdly, the condition in (4) is "pathological" in the distributed setting as it restricts the heterogeneity between different clients and can result in vacuous bounds (Khaled et al., 2020).

Therefore, to enforce bounded sensitivity in practice (Abadi et al., 2016), it is recommended to use clipping with threshold $\tau > 0$

$$\text{Clip}_\tau(g) := \min\left(1, \frac{\tau}{\|g\|}\right) g. \quad (5)$$

In this case, the sensitivity $\Psi$ is bounded above by the clipping threshold $\tau$, as $\|\Psi(g)\| = \|\text{Clip}_\tau(g)\| \leq \tau = \Phi$. In fact, the method (2) that uses clipping (5) is typically referred to as DP-SGD in the literature. It was analyzed under the symmetric noise assumption by Chen et al. (2020). However, Koloskova et al. (2023) showed that without additional restrictive assumptions, DP-SGD even in the absence of DP noise does not converge due to the bias introduced by clipping operator (5). Furthermore, as large values of $\tau$ imply stronger privacy, jointly optimizing convergence and privacy of DP-SGD by carefully tuning $\tau$ and $\gamma$ in the DP setting is a challenging task (Kurakin et al., 2022; Bu et al., 2024).

**Smoothed normalization as an alternative to clipping.** To eliminate the need to tune the threshold $\tau$ of clipping, smoothed normalization is an alternative operator (Bu et al., 2024; Yang et al., 2022) with its parameter that provides robust convergence performance of DP-SGD. The operator is defined by

$$\text{Norm}_\alpha(g) := \frac{1}{\alpha + \|g\|} g, \quad (6)$$

for some $\alpha \geq 0$ and satisfies the following property.

**Lemma 1.** *For any $\alpha \geq 0$, $\beta > 0$, and $g \in \mathbb{R}^d$,*

$$\|\text{Norm}_\alpha(g)\| \leq 1, \quad (7)$$

$$\|g - \beta\text{Norm}_\alpha(g)\|^2 = \left(1 - \frac{\beta}{\alpha + \|g\|}\right)^2 \|g\|^2. \quad (8)$$

Clearly, smoothed normalization ensures Property (7) that the norm of the normalized vector is bounded above by 1. Also, Property (8) states that the distance between the true vector and a $\beta$-multiple of the normalized vector is bounded by a function of $\beta$, $\alpha$, and $\|g\|$. Furthermore, note that smoothed normalization with $\alpha = 0$ recovers standard normalization $g/\|g\|$ by Nesterov (1984); Hazan et al. (2015); Levy (2016). However, smoothed normalization with $\alpha > 0$ helps improve the contraction factor, compared to standard normalization. Specifically, as $\|g\| \to 0$, the contraction factor of smoothed normalization approaches $(1 - \beta/\alpha)^2$. However, standard normalization lacks this contraction property.

DP-SGD in (2) with smoothed normalization achieves robust empirical convergence in the DP setting (Bu et al., 2024). Nonetheless, the convergence of this method in the single-node setting without the bounded gradient norm assumption by Bu et al. (2024) still depends on the central symmetry of stochastic gradients around the true gradient.

### 3.4. Limitations of DP Distributed Gradient Methods

Extending the convergence results of DP-SGD to the distributed setting poses significant challenges due to potential client heterogeneity. Existing results often address the bias introduced by the operator (clipping or normalization) by relying on restrictive assumptions, such as assuming that clipping is effectively turned off (Zhang et al., 2024; Noble et al., 2022), or imposing boundedness of gradient norms (Li et al., 2022; Zhang et al., 2022; Murata & Suzuki, 2023; Wang et al., 2024). A recent work by Li et al. (2024) extended the analysis of Koloskova et al. (2023) to a distributed private setting under strong gradient dissimilarity condition. However, their method fails to converge due to the limitation of clipping, as discussed earlier. More importantly, even in the absence of the DP noise ($z^k = 0$), the inherent bias in the gradient estimator can severely impact the convergence. For instance, the methods with update (2) can diverge exponentially when $\Psi(\cdot)$ is a Top-1 compressor (Beznosikov et al., 2023), and fail to converge when $\Psi(\cdot)$ is a clipping operator (Chen et al., 2020; Khirirat et al., 2023). Moreover, smoothed normalization (6) with $\alpha = 0$ also cannot address this problem as demonstrated in the following example.

**Example 1.** *Consider Problem (1) with $n = 2, d = 1$, $f_1(x) = \frac{1}{2}(x-3)^2$ and $f_2(x) = \frac{1}{2}(x+3)^2$. Then $f(x) = \frac{1}{2}(f_1(x) + f_2(x))$ is minimized at $x^\star = 0$ and satisfies Assumption 1. The iterates $\{x^k\}$ generated by (2)*

*(for $B = 2$) with $z^k = 0$ and $\alpha = 0$ do not progress when $x^0 = 2$, as the gradient estimator* $\text{Norm}_\alpha\left(\nabla f_1(x^k)\right) + \text{Norm}_\alpha\left(\nabla f_2(x^k)\right)$ *results in*

$$\frac{\nabla f_1(x^0)}{\|\nabla f_1(x^0)\|} + \frac{\nabla f_2(x^0)}{\|\nabla f_2(x^0)\|} = -1/1 + 5/5 = 0.$$

Thus, applying normalization directly to the gradients in DP-SGD leads to the method that does not converge in the distributed setting without additional assumptions. Moreover, Example 1 shows a fundamental limitation of algorithms relying on normalization of the client updates (Das et al., 2021).

### 3.5. EF21 Mechanism

To resolve the convergence issues of distributed gradient methods with biased operators, one approach is to use EF21, an error feedback mechanism developed by Richtárik et al. (2021). Instead of directly applying the biased gradient estimator $\Psi$ to the gradient, EF21 applies $\Psi$ to the *difference* between the true gradient and the current error feedback vector. At each iteration of the modified method $k = 0, 1, \ldots, K$, each client $i$ receives the current iterate $x^k$ from the central server, and computes its local update $g_i^{k+1}$ via

$$g_i^{k+1} = g_i^k + \beta\Psi(\nabla f_i(x^k) - g_i^k), \tag{9}$$

where $\beta > 0$. Next, the central server receives the average of local error-feedback vectors that are communicated by all clients $\frac{1}{n}\sum_{i=1}^n \Psi(\nabla f_i(x^k) - g_i^k)$, computes the global gradient estimator $g^k := \frac{1}{n}\sum_{i=1}^n g_i^k$ as

$$g^{k+1} = g^k + \frac{\beta}{n}\sum_{i=1}^n \Psi(\nabla f_i(x^k) - g_i^k), \tag{10}$$

and updates the next iterate $x^{k+1}$ via

$$x^{k+1} = x^k - \gamma g^{k+1}. \tag{11}$$

This method generalizes EF21, which utilizes a contractive compressor (Stich et al., 2018; Beznosikov et al., 2023) is defined by

$$\|g - \mathcal{C}(g)\|^2 \le (1 - \eta)^2 \|g\|^2,$$

for some $\eta \in (0, 1]$ and any $g \in \mathbb{R}^d$. Rather, the method encompasses other estimators $\Psi(\cdot)$ such as clipping in Clip21 proposed by Khirirat et al. (2023).

Despite achieving the $\mathcal{O}(1/K)$ convergence in the non-private setting, Clip21 faces difficulty in establishing provable convergence in the presence of DP noise. First, its convergence analysis relies on descent inequalities that separately consider cases where clipping is active and inactive, as the clipping operator does not satisfy the contractive compressor property required by EF21 (see Table 1). Second,

the clipping threshold $\tau$ intricately influences both privacy and convergence. To obtain the descent inequality, $\tau$ has to be chosen sufficiently high, which leads to adding large Gaussian noise. The accumulation of the DP noise prevents the convergence. These properties of clipping make it challenging to establish convergence guarantees for Clip21 in the DP setting.

## 4. $\alpha$-Norm21 in the Non-Private Setting

To address the convergence challenges of Clip21, we propose $\alpha$-NormEC, the first distributed method to provide provable convergence guarantees in the DP setting. $\alpha$-NormEC implements the update rules defined by (9), (10), and (11), where $\Psi(\cdot)$ is smoothed normalization (6) that offers key advantages over clipping. In the update rule in (11), we use server normalization $x^{k+1} = x^k - \gamma g^{k+1}/\|g^{k+1}\|$ and adopt notation $0/0 = 0$. See Algorithm 1 for the detailed description of $\alpha$-NormEC.

---

**Algorithm 1** (DP-)$\alpha$-NormEC

1: **Input:** Step size $\gamma > 0$; $\beta > 0$; normalization parameter $\alpha > 0$; starting points $x^0, g_i^0 \in \mathbb{R}^d$ for $i \in [1, n]$ and $\hat{g}^0 = \frac{1}{n}\sum_{i=1}^n g_i^0$; $z_i^k \in \mathbb{R}^d$ are sampled from Gaussian distribution with zero mean and $\sigma_{\text{DP}}^2$-variance.
2: **for** each iteration $k = 0, 1, \ldots, K$ **do**
3:     **for** each client $i = 1, 2, \ldots, n$ in parallel **do**
4:         Compute local gradient $\nabla f_i(x^k)$
5:         Compute $\Delta_i^k = \text{Norm}_\alpha\left(\nabla f_i(x^k) - g_i^k\right)$
6:         Update $g_i^{k+1} = g_i^k + \beta\Delta_i^k$
7:         **Non-private setting:** Transmit $\hat{\Delta}_i^k = \Delta_i^k$
8:         **Private setting:** Transmit $\hat{\Delta}_i^k = \Delta_i^k + z_i^k$
9:     **end for**
10:    Server computes $\hat{g}^{k+1} = \hat{g}^k + \frac{\beta}{n}\sum_{i=1}^n \hat{\Delta}_i^k$
11:    Server updates $x^{k+1} = x^k - \gamma\hat{g}^{k+1}/\|\hat{g}^{k+1}\|$
12: **end for**
13: **Output:** $x^{K+1}$

---

We show that $\alpha$-NormEC provides stronger convergence guarantees than Clip21 in the non-private setting, and achieves the first convergence guarantees in the DP setting. These theoretical benefits of $\alpha$-NormEC stem from favorable properties of smoothed normalization. Specifically, smoothed normalization, unlike clipping, behaves similarly to a contractive compressor (see Table 1), which simplifies the convergence analysis of $\alpha$-NormEC compared to Clip21. Furthermore, the smoothed normalization parameter, unlike the clipping threshold, does not affect the DP noise variance, thus facilitating the extension to the DP setting while maintaining robust convergence.

Now, we begin by presenting the convergence results of $\alpha$-NormEC in the non-private setting.

**Theorem 1.** *Consider Algorithm 1 for solving Problem (1)*

| Operator | Property |
|---|---|
| Contractive compressor $\mathcal{C} : \mathbb{R}^d \to \mathbb{R}^d$ | $\|\mathcal{C}(g) - g\|^2 \leq (1-\eta)^2 \|g\|^2$ |
| Clipping $\text{Clip}_\tau(g) := \min\left(1, \frac{\tau}{\|g\|}\right) g$ | $\|\text{Clip}_\tau(g) - g\|^2 \leq \max(0, \|g\| - \tau)^2$ |
| Smoothed normalization $\text{Norm}_\alpha(g) := \frac{1}{\alpha + \|g\|} g$ | $\|\text{Norm}_\alpha(g) - g\|^2 \leq \left(1 - \frac{1}{\alpha + \|g\|}\right)^2 \|g\|^2$ |

Table 1: Comparisons of the property of contractive compressor, clipping, and smoothed normalization. Unlike clipping, smoothed normalization obtains the contractive property similar to contractive compressors.

*in the non-private setting, where Assumption 1 holds. Let $\beta, \alpha, \gamma > 0$ be chosen such that*

$$\frac{\beta}{\alpha + R} < 1, \quad and \quad \gamma \leq \frac{\beta R}{\alpha + R} \frac{1}{L_{\max}},$$

*where $R = \max_{i \in [1,n]} \left\|\nabla f_i(x^0) - g_i^0\right\|$ and $L_{\max} = \max_{i \in [1,n]} L_i$. Then,*

$$\min_{k \in [0,K]} \left\|\nabla f(x^k)\right\| \leq \frac{f(x^0) - f^{\inf}}{\gamma(K+1)} + 2R + \frac{L}{2}\gamma.$$

Theorem 1 demonstrates that in the non-private setting, $\alpha$-NormEC converges sublinearly up to the additive constant of $2R + \frac{L}{2}\gamma$. This constant diminishes when we properly choose initialized memory vectors $g_i^{-1}$ and reduce the stepsize $\gamma$, as shown in the next corollary.

**Corollary 1.** *Consider Algorithm 1 for solving Problem (1) under the same setting as Theorem 1. If we choose $g_i^0 \in \mathbb{R}^d$ such that $\max_{i \in [1,n]} \left\|\nabla f_i(x^0) - g_i^0\right\| = \frac{D}{(K+1)^{1/2}}$ with any $D > 0$, $\gamma \leq \frac{\beta}{L_{\max}} \frac{D}{\alpha + D} \frac{1}{(K+1)^{1/2}}$, and $\alpha > \beta$, then*

$$\min_{k \in [0,K]} \left\|\nabla f(x^k)\right\| \leq \frac{C}{(K+1)^{1/2}},$$

*where $C = \frac{L_{\max}(\alpha + D)}{\beta D}(f(x^0) - f^{\inf}) + 2D + \frac{L}{2}\frac{\beta D}{L_{\max}(\alpha + D)}$.*

According to Corollary 1, $\alpha$-NormEC enjoys the $\mathcal{O}(1/\sqrt{K})$ convergence rate in the gradient norm when we choose $g_i^{-1}$ such that $R = \mathcal{O}(1/\sqrt{K})$ and $\gamma = \mathcal{O}(\beta/\sqrt{K})$. By further choosing $\alpha > 1$, and

$$\beta = \frac{L_{\max}(\alpha + D)}{D}\sqrt{\frac{2(f(x^0) - f^{\inf})}{L}},$$

which ensures $\frac{L_{\max}(\alpha + D)}{\beta D}(f(x^0) - f^{\inf}) = \frac{L}{2}\frac{\beta D}{L_{\max}(\alpha + D)}$, the associated convergence bound from Corollary 1 becomes

$$\min_{k \in [0,K]} \left\|\nabla f(x^k)\right\| \leq \frac{\sqrt{2L(f(x^0) - f^{\inf})} + 2D}{(K+1)^{1/2}}. \quad (12)$$

This convergence bound (12) comprises two terms. The $\frac{\sqrt{2L(f(x^0) - f^{\inf})}}{(K+1)^{1/2}}$-term is the convergence bound obtained by

classical gradient descent, while the $\frac{2D}{(K+1)^{1/2}}$-term comes from the initialized memory vectors $g_i^{-1}$ for running the error-feedback mechanism.

**Comparison between $\alpha$-NormEC and Clip21.** In the non-private setting, $\alpha$-NormEC provides stronger convergence guarantees than Clip21. First, the hyperparameters of $\alpha$-NormEC ($\beta, \alpha, \gamma > 0$), as defined in Theorem 1, are easy to implement. Conversely, the stepsize $\gamma$ of Clip21 (Theorem 5.6 of Khirirat et al. (2019)) presents a practical challenge, as it depends on the inaccessible values of $f(x^0) - f^{\inf}$. Furthermore, the convergence bound of $\alpha$-NormEC (12) exhibits a smaller convergence factor than that of Clip21, as detailed in Appendix E. Specifically, by choosing $g_i^0 \in \mathbb{R}^d$ such that $D$ is sufficiently small, the convergence bound of $\alpha$-NormEC in (12) approaches that of classical gradient descent (Carmon et al., 2020).

**Proof outline of $\alpha$-NormEC.** We outline the proof for $\alpha$-NormEC. By the $L$-smoothness of the objective function $f$, and by the update for $x^{k+1}$ in $\alpha$-NormEC,

$$V^{k+1} \leq V^k - \gamma \left\|\nabla f(x^k)\right\| + \frac{L\gamma^2}{2} + 2\gamma W^k,$$

where $V^k := f(x^k) - f^{\inf}$, and $W^k := \frac{1}{n}\sum_{i=1}^n \left\|\nabla f_i(x^k) - g_i^{k+1}\right\|$. The key step to establish the convergence is to bound $\left\|\nabla f_i(x^k) - g_i^{k+1}\right\|$. From Lemma 2, with appropriate choices of the tuning parameters $\beta$, $\alpha$, and $\gamma$, we obtain

$$\left\|\nabla f_i(x^k) - g_i^{k+1}\right\| \leq \max_{i \in [1,n]} \left\|\nabla f_i(x^0) - g_i^0\right\|, \quad \forall k \geq 0.$$

Finally, substituting this bound into the previous inequality yields the convergence bound in $\min_{k \in [0,K]} \left\|\nabla f(x^k)\right\|$. Deriving the bound on $\left\|\nabla f_i(x^k) - g_i^{k+1}\right\|$ for $\alpha$-NormEC by induction is similar to but simpler than Clip21. This simplified proof is possible, because smoothed normalization possesses a contractive property similar to the contractive compressor used in EF21.

## 5. $\alpha$-Norm21 in the DP Setting

Next, we extend $\alpha$-NormEC to the DP setting. The DP version of $\alpha$-NormEC is identical to its non-private counterpart, except for the step of communicating $\hat{\Delta}_i^k$ of Algorithm 1. In this step, instead of transmitting the non-private normalized gradient $\hat{\Delta}_i^k = \Delta_i^k := \text{Norm}_\alpha \left( \nabla f_i(x^k) - g_i^k \right)$ as done in the non-private version, each client in the DP version communicates the DP normalized gradient $\hat{\Delta}_i^k = \Delta_i^k + z_i^k$, where $z_i^k$ is the DP noise.

The next theorem presents the convergence rate for $\alpha$-NormEC in the DP setting.

**Theorem 2.** *Consider Algorithm 1 for solving Problem* (1) *in the private setting, where Assumption 1 holds. Let* $\beta, \alpha, \gamma > 0$ *be chosen such that*

$$\frac{\beta}{\alpha + R} < 1, \quad and \quad \gamma \le \frac{\beta R}{\alpha + R} \frac{1}{L_{\max}},$$

*where* $R = \max_{i \in [1,n]} \left\| \nabla f_i(x^0) - g_i^0 \right\|$, *and* $L_{\max} = \max_{i \in [1,n]} L_i$. *Then,*

$$\min_{k \in [0,K]} \mathrm{E} \left[ \left\| \nabla f(x^k) \right\| \right] \le \frac{f(x^0) - f^{\inf}}{\gamma(K+1)} + 2R + \frac{L}{2}\gamma$$
$$+ 2\sqrt{\beta^2(K+1)\sigma_{\mathrm{DP}}^2}.$$

In the DP setting, from Theorem 2, $\alpha$-NormEC achieves the sublinear convergence up to the additive constant of $2R + \frac{L}{2}\gamma + 2\sqrt{\beta^2(K+1)\sigma_{\mathrm{DP}}^2}$. Notice that $\alpha$-NormEC in the DP setting introduces one additional constant that arises from the DP noise $\sigma_{\mathrm{DP}}^2$. This additive constant diminishes, when we choose initialized memory vectors $g_i^0 \in \mathbb{R}^d$ such that $R$ becomes small, and decrease tuning parameters $\gamma, \beta > 0$.

**Utility guarantees.** In the DP setting, unlike Clip21 (Khirirat et al., 2023), $\alpha$-NormEC achieves the $(\epsilon, \delta)$-DP, and obtains the utility-privacy trade-off. We show this by setting the standard deviation of the DP noise according to Theorem 1 of Abadi et al. (2016), i.e. $\sigma_{\mathrm{DP}} = \mathcal{O}(\sqrt{(K+1)\log(1/\delta)}\epsilon^{-1})$, which yields the following utility bound.

**Corollary 2** (Utility guarantee). *Consider Algorithm 1 for solving Problem* (1) *under the same setting as Theorem 2. If* $\sigma_{\mathrm{DP}} = \mathcal{O}(\sqrt{(K+1)\log(1/\delta)}\epsilon^{-1})$, *and* $\beta = \frac{\beta_0}{K+1}$ *with* $\beta_0 \le \Delta \sqrt[4]{n\epsilon^2/(d\log(1/\delta))}$ *and* $\alpha > \beta_0$, *then Algorithm 1 satisfies* $(\epsilon, \delta)$-*DP while attaining the bound:*

$$\min_{k \in [0,K]} \mathrm{E} \left[ \left\| \nabla f(x^k) \right\| \right] \le \mathcal{O} \left( \Delta \sqrt[4]{\frac{d\log(1/\delta)}{n\epsilon^2}} \right) + 2R,$$

*where* $\Delta = \sqrt{L_{\max}(\alpha + R)(f(x^0) - f^{\inf})/R}$, *and* $R = \max_{i \in [1,n]} \left\| \nabla f_i(x^0) - g_i^0 \right\|$.

Unlike Clip21, $\alpha$-NormEC provides the first utility bound in the DP distributed setting that accounts for the effect of smoothed normalization, a factor often neglected in existing literature. As $R$ is sufficiently small ($R \to 0$), $\alpha$-NormEC achieves the utility bound of $\mathcal{O} \left( \Delta \sqrt[4]{\frac{d\log(1/\delta)}{n\epsilon^2}} \right)$. Our obtained utility bound applies for smooth problems without the bounded gradient norm assumption, the limitation present in prior work that analyzes DP-SGD such as Li et al. (2022); Wang et al. (2023); Lowy et al. (2023); Zhang et al. (2020).

## 6. Experiments

We present the numerical evaluation of $\alpha$-NormEC by solving a non-convex optimization problem of training deep neural networks. We consider the image classification task with the CIFAR-10 (Krizhevsky et al., 2009) dataset using the ResNet20 (He et al., 2016) model. Experimental details are provided in the Appendix H.

**Sensitivity of $\alpha$-NormEC to hyper-parameters.** We investigate the impact of hyperparameters $\alpha$ and $\beta$ on the performance of $\alpha$-NormEC in the non-private training. Figure 1 visualizes the highest test accuracy achieved during training over 300 communication rounds with a fine-tuned, constant step size $\gamma$, while we vary $\beta$ and $\alpha$. Appendix H.1 presents additional metrics and convergence curves.

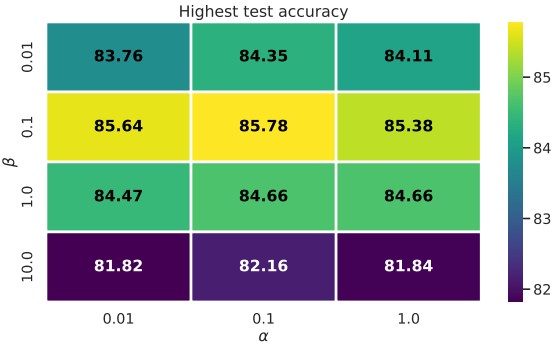

Figure 1: Best test accuracy achieved by $\alpha$-NormEC.

Figure 1 reveals that in the non-private training, the convergence of $\alpha$-NormEC is stable with respect to a wide range of $\alpha$ values and robust to $\beta$. The performance of $\alpha$-NormEC is primarily governed by the choice of $\beta$. Optimal performance (85-86% accuracy) is observed when $\beta$ is around 0.1. While $\alpha$-NormEC is stable with respect to $\alpha$, extreme values of $\beta$ lead to suboptimal performance: very large values ($\beta = 10.0$) result in significantly lower accuracy (81-82%), while very small values ($\beta = 0.01$) achieve moderate performance (83-84%). The optimal configuration, achieving the highest 85.78% accuracy, is $\beta = 0.1$ and $\alpha = 0.1$. For further experiments, we adopt $\alpha = 0.01$, aligning with recommendations from prior empirical works

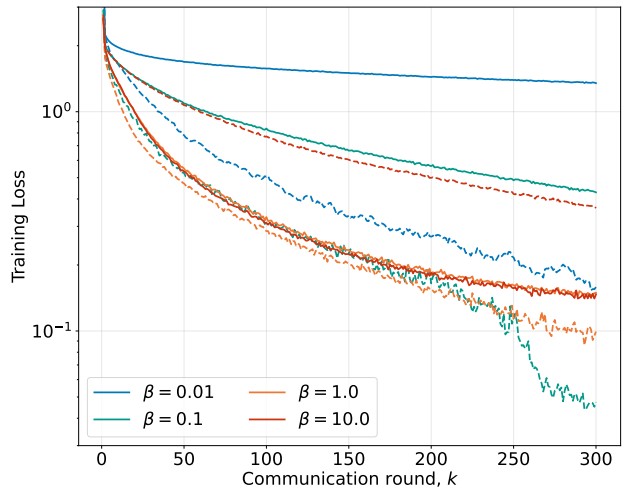

Figure 2: Comparison of DP-SGD (2) [solid] and $\alpha$-NormEC (1) [dashed] without server normalization.

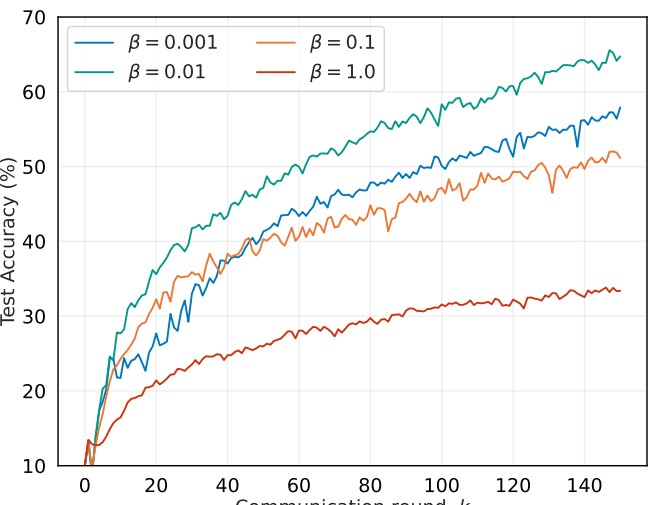

Figure 3: Performance of DP-$\alpha$-NormEC.

in the single-node setting (Bu et al., 2024).

**Effect of Error Compensation (EC).** We examine how EC improves the convergence performance of distributed gradient methods using smoothed normalization in non-private training. To isolate the effect of EC, we compare $\alpha$-NormEC 1 without server normalization (Line 11) to a DP-SGD method (with smoothed normalization) governed by Equation (2) with $B = n, z \equiv 0$. Figure 2 displays convergence in training loss across different $\beta$ (with tuned step size $\gamma$). In Appendix H.2, we also report the behavior of test accuracy in Figure 8 and optimal parameters with final accuracies in Figure 9.

Figure 2 demonstrates the substantial convergence improvements achieved by EC for distributed gradient methods with smoothed normalization across most $\beta$ values, with the exception of $\beta = 10$. This large $\beta$ value, however, is impractical for differentially private settings due to increased noise variance. Moreover, while $\alpha$-NormEC exhibits robust performance across different $\beta$ values, DP-SGD shows higher sensitivity to this parameter choice, particularly struggling with convergence when $\beta = 0.01$. This comparison highlights how EC not only improves convergence but also enhances the algorithm's stability across different parameter settings.

Furthermore, we present an ablation study on the effect of server normalization in Appendix H.3. Due to space constraints the comparison between $\alpha$-NormEC and Clip21 is presented in Appendix H.4.

**Private training.** We analyze the performance of $\alpha$-NormEC in the differentially private setting. We set the noise variance at $\beta\sqrt{K \log(1/\delta)}\epsilon^{-1}$ for $\epsilon = 8, \delta = 10^{-5}$. The test accuracy results in Figure 3 demonstrate that $\alpha$-NormEC's performance is highly dependent on the choice

of parameter $\beta$. Small values ($\beta = 0.01$) achieve the best performance, reaching approximately 65% accuracy, while maintaining stable convergence throughout training. Moderate values ($\beta = 0.1$) show slightly slower convergence but eventually reach similar performance levels. However, larger values ($\beta = 1.0$) significantly degrade the performance, with $\beta = 1.0$ barely exceeding 33% accuracy due to excessive noise injection required for privacy guarantees.

## 7. Conclusion

We have proposed and analyzed $\alpha$-NormEC, a novel distributed algorithm that integrates smoothed normalization with the EF21 mechanism for solving non-convex, smooth optimization problems in both non-private and private settings. Unlike Clip21, $\alpha$-NormEC achieves strong convergence guarantees that almost match those of classical gradient descent for non-private training, and provides the first utility bound for private training without relying on restrictive assumptions such as bounded gradient norms. Our experiments on neural network training demonstrate that the proposed method achieves robust convergence performance with respect to its parameters. Moreover, $\alpha$-NormEC significantly outperforms distributed gradient methods with direct smoothed normalization in terms of accuracy.

**Future work.** Our work implies many promising research directions. One direction is to extend $\alpha$-NormEC to accommodate the partial participation case, where the central server receives the local normalized gradients from a few clients, and the stochastic case, where each client has access only to stochastic gradients. Another important direction is to modify $\alpha$-NormEC to solve federated learning problems, where the clients run their local updates before the local updates are normalized and transmitted to the central server.

## Impact Statement

This paper proposes distributed optimization methods for machine learning and differential privacy. Unlike existing literature, our proposed methods are more practical for deployment in both non-private and private training, offering strong convergence guarantees and, for the first time, utility guarantees under a specified privacy budget. Additionally, the hyperparameters of the proposed methods are straightforward to implement, enhancing their practicality for real-world FL applications.

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

# Contents

# A. Proof of Lemma 1

We prove the first statement by taking the Euclidean norm. Next, we prove the second statement. From the definition of the Euclidean norm,

$$\left\|g - \beta\mathrm{Norm}_\alpha\left(g\right)\right\|^2 \overset{(6)}{=} \|g\|^2 + \frac{\beta^2}{(\alpha + \|g\|)^2}\|g\|^2 - 2\beta\frac{\|g\|^2}{\alpha + \|g\|}$$

$$= \left(1 - \frac{\beta}{\alpha + \|g\|}\right)^2\|g\|^2.$$

# B. Comparison of EF21 between Clipping and Smoothed Normalization

In this section, we compare the EF21 mechanism that is modified by replacing a contractive compressor with clipping in Clip21, and with smoothed normalization in $\alpha$-NormEC. To compare these modified updates, given the optimal vector $g^\star \in \mathbb{R}^d$, consider the single-node EF21 mechanism, which computes the memory vector $g^k \in \mathbb{R}^d$ according to

$$g^{k+1} = g^k + \Psi(g^\star - g^k), \tag{13}$$

where $\Psi : \mathbb{R}^d \to \mathbb{R}^d$ is the biased gradient estimator and $g^0 \in \mathbb{R}^d$ is the initial memory vector.

If $\Psi(g) = \mathrm{Clip}_\tau\left(g\right)$, then from Theorem 4.3 of Khirirat et al. (2023)

$$\left\|g^k - g^\star\right\| \leq \max(0, \left\|g^0 - g^\star\right\| - k\tau).$$

If $\Psi(g) = \mathrm{Norm}_\alpha\left(g\right)$, then from Lemma 1

$$\left\|g^\star - g^k\right\|^2 = \left\|g^\star - g^{k-1} - \beta\mathrm{Norm}_\alpha\left(g^\star - g^{k-1}\right)\right\|^2$$

$$= \left(1 - \frac{\beta}{\alpha + \|g^\star - g^{k-1}\|}\right)^2\left\|g^\star - g^{k-1}\right\|^2$$

$$\vdots$$

$$= \left\|g^\star - g^0\right\|^2 \cdot \prod_{l=1}^{k}\left(1 - \frac{\beta}{\alpha + \|g^\star - g^{l-1}\|}\right)^2.$$

In conclusion, while the EF21 mechanism with clipping ensures that the memory $g^k$ will reach $g^\star$ within a finite number of iterations $k$ (when $k \geq \left\|g^0 - g^\star\right\|/\tau$), the EF21 mechanism with smoothed normalization guarantees that $g^k$ will eventually reach $g^\star$ (provided that $\beta/\alpha < 1$).

# C. Proof of Theorem 1

To prove the result in Theorem 1 requires us to utilize the following lemma, which shows $\left\|\nabla f_i(x^{k+1}) - g_i^{k+1}\right\| \leq R$ for some positive scalars $R$, given that $\left\|\nabla f_i(x^k) - g_i^k\right\| \leq R$.

**Lemma 2.** *Consider Algorithm 1 for solving Problem (1) in the non-private setting, where Assumption 1 holds. If $\left\|\nabla f_i(x^k) - g_i^k\right\| \leq R$, $\frac{\beta}{\alpha+R} < 1$, and $\gamma \leq \frac{\beta R}{\alpha+R}\frac{1}{L_{\max}}$ with $L_{\max} = \max_{i\in[1,n]} L_i$, then $\left\|\nabla f_i(x^{k+1}) - g_i^{k+1}\right\| \leq R$.*

*Proof.* From the definition of the Euclidean norm,

$$\left\|\nabla f_i(x^{k+1}) - g_i^{k+1}\right\| \overset{\text{triangle inequality}}{\leq} \left\|\nabla f_i(x^{k+1}) - \nabla f_i(x^k)\right\| + \left\|\nabla f_i(x^k) - g_i^{k+1}\right\|$$

$$\overset{g_i^{k+1}}{=} \left\|\nabla f_i(x^{k+1}) - \nabla f_i(x^k)\right\| + \left\|\nabla f_i(x^k) - g_i^k - \beta\mathrm{Norm}_\alpha\left(\nabla f_i(x^k) - g_i^k\right)\right\|$$

$$\overset{\text{Lemma 1}}{\leq} \left\|\nabla f_i(x^{k+1}) - \nabla f_i(x^k)\right\| + \left|1 - \frac{\beta}{\alpha + \left\|\nabla f_i(x^k) - g_i^k\right\|}\right|\left\|\nabla f_i(x^k) - g_i^k\right\|$$

$$\overset{\text{Assumption 1, and } x^{k+1}}{\leq} L_{\max}\gamma + \left|1 - \frac{\beta}{\alpha + \left\|\nabla f_i(x^k) - g_i^k\right\|}\right|\left\|\nabla f_i(x^k) - g_i^k\right\|.$$

If $\left\|\nabla f_i(x^k) - g_i^k\right\| \leq R$, and $\frac{\beta}{\alpha+R} < 1$, then $\left\|\nabla f_i(x^{k+1}) - g_i^{k+1}\right\| \leq R$ when

$$\gamma \leq \frac{\beta R}{\alpha + R} \frac{1}{L_{\max}}.$$

$\square$

Now, we are ready to prove the result in Theorem 1 in four steps.

**Step 1) Prove by induction that** $\left\|\nabla f_i(x^k) - g_i^k\right\| \leq R$ **for** $R = \max_{i \in [1,n]} \left\|\nabla f_i(x^0) - g_i^0\right\|$. For $k = 0$, this is obvious. Next, let $\left\|\nabla f_i(x^l) - g_i^l\right\| \leq R$ for $R = \max_{i \in [1,n]} \left\|\nabla f_i(x^0) - g_i^0\right\|$ for $l = 0, 1, \ldots, k$. Then, if $\beta/(\alpha + R) < 1$, and $\gamma \leq \frac{\beta R}{\alpha+R} \frac{1}{L_{\max}}$, then from Lemma 2 $\left\|\nabla f_i(x^{k+1}) - g_i^{k+1}\right\| \leq R$.

**Step 2) Bound** $\left\|\nabla f_i(x^k) - g_i^{k+1}\right\|$. From the definition of the Euclidean norm,

$$
\begin{aligned}
\left\|\nabla f_i(x^k) - g_i^{k+1}\right\| & \overset{g_i^{k+1}}{=} \left\|\nabla f_i(x^k) - g_i^k - \beta \mathrm{Norm}_\alpha\left(\nabla f_i(x^k) - g_i^k\right)\right\| \\
& \overset{\text{Lemma } 1}{\leq} \left|1 - \frac{\beta}{\alpha + \left\|\nabla f_i(x^k) - g_i^k\right\|}\right| \left\|\nabla f_i(x^k) - g_i^k\right\| \\
& \overset{\beta/(\alpha+R)<1}{\leq} \left(1 - \frac{\beta}{\alpha + R}\right) R \leq R.
\end{aligned}
$$

**Step 3) Derive the descent inequality.** By the $L$-smoothness of $f$, by the definition of $x^{k+1}$, and by the fact that $\hat{g}^{k+1} = g^{k+1}$,

$$
\begin{aligned}
f(x^{k+1}) - f^{\inf} & \leq f(x^k) - f^{\inf} - \frac{\gamma}{\|g^{k+1}\|} \left\langle \nabla f(x^k), g^{k+1}\right\rangle + \frac{L\gamma^2}{2} \\
& = f(x^k) - f^{\inf} - \gamma\left\|g^{k+1}\right\| + \frac{\gamma}{\|g^{k+1}\|} \left\langle \nabla f(x^k) - g^{k+1}, g^{k+1}\right\rangle + \frac{L\gamma^2}{2} \\
& \overset{\text{Cauchy-Schwartz inequality}}{\leq} f(x^k) - f^{\inf} - \gamma\left\|g^{k+1}\right\| + \gamma\left\|\nabla f(x^k) - g^{k+1}\right\| + \frac{L\gamma^2}{2} \\
& \overset{\text{triangle inequality}}{\leq} f(x^k) - f^{\inf} - \gamma\left\|\nabla f(x^k)\right\| + 2\gamma\left\|\nabla f(x^k) - g^{k+1}\right\| + \frac{L\gamma^2}{2} \\
& \overset{\text{triangle inequality}}{\leq} f(x^k) - f^{\inf} - \gamma\left\|\nabla f(x^k)\right\| + 2\gamma\frac{1}{n}\sum_{i=1}^{n}\left\|\nabla f_i(x^k) - g_i^{k+1}\right\| + \frac{L\gamma^2}{2}.
\end{aligned}
$$

Since $\left\|\nabla f_i(x^k) - g_i^{k+1}\right\| \leq R$ with $R = \max_{i \in [1,n]}\left\|\nabla f_i(x^0) - g_i^0\right\|$, we have

$$f(x^{k+1}) - f^{\inf} \leq f(x^k) - f^{\inf} - \gamma\left\|\nabla f(x^k)\right\| + 2\gamma \max_{i \in [1,n]}\left\|\nabla f_i(x^0) - g_i^0\right\| + \frac{L\gamma^2}{2}.$$

**Step 4) Finalize the convergence rate.** Now, we prove the first statement. By re-arranging the terms of the inequality,

$$
\begin{aligned}
\min_{k \in [0,K]}\left\|\nabla f(x^k)\right\| & \leq \frac{1}{K+1}\sum_{k=0}^{K}\left\|\nabla f(x^k)\right\| \\
& \leq \frac{[f(x^0) - f^{\inf}] - [f(x^{K+1}) - f^{\inf}]}{\gamma(K+1)} + 2\max_{i \in [1,n]}\left\|\nabla f_i(x^0) - g_i^0\right\| + \frac{L}{2}\gamma \\
& \overset{f^{\inf} \geq f(x^{K+1})}{\leq} \frac{f(x^0) - f^{\inf}}{\gamma(K+1)} + 2\max_{i \in [1,n]}\left\|\nabla f_i(x^0) - g_i^0\right\| + \frac{L}{2}\gamma.
\end{aligned}
$$

## D. Proof of Corollary 1

If $g_i^0 \in \mathbb{R}^d$ is chosen such that $\max_{i \in [1,n]} \left\| \nabla f_i(x^0) - g_i^0 \right\| = \frac{D}{(K+1)^{1/2}}$ with any $D > 0$, $\gamma \le \frac{\beta}{L_{\max}} \frac{D}{\alpha+D} \frac{1}{(K+1)^{1/2}}$, and $\beta < \alpha$, then $\gamma \le \frac{\beta R}{\alpha+R} \frac{1}{L_{\max}}$ with $R = \max_{i \in [1,n]} \left\| \nabla f_i(x^0) - g_i^0 \right\|$, and thus

$$\min_{k \in [0,K]} \left\| \nabla f(x^k) \right\| \le \frac{L_{\max}(\alpha+D)}{\beta D} \frac{f(x^0) - f^{\inf}}{(K+1)^{1/2}} + 2\frac{D}{(K+1)^{1/2}} + \frac{L}{2} \frac{\beta D}{L_{\max}(\alpha+D)} \frac{1}{(K+1)^{1/2}}.$$

## E. $\alpha$-NormEC and Clip21 Comparison

We compare the convergence bound of $\alpha$-NormEC in (12) with Clip21 (Khirirat et al., 2023). In particular, the convergence factor of $\alpha$-NormEC in (12) is potentially smaller than that of Clip21 from Theorem 5.6. of Khirirat et al. (2023)

Let $\hat{x}^K$ be selected uniformly at random from a set $\{x^0, x^1, \dots, x^K\}$. Then, from Theorem 5.6. of Khirirat et al. (2023), Clip21 converges at the rate:

$$\min_{k \in [0,K]} \left\| \nabla f(x^k) \right\| \le \mathrm{E}\left[ \left\| \nabla f(\hat{x}^K) \right\| \right]$$

$$\le \sqrt{\mathrm{E}\left[ \left\| \nabla f(\hat{x}^K) \right\|^2 \right]}$$

$$\le \frac{L_{\max}(f(x^0) - f^{\inf})}{\tau(K+1)^{1/2}} + \frac{\sqrt{(1 + C_1/\tau)C_2}}{(K+1)^{1/2}},$$

where $\tau > 0$ is a clipping threshold, $C_1 = \max_{i \in [1,n]} \left\| \nabla f_i(x^0) \right\|$, and $C_2 = \max(\max(L, L_{\max})(f(x^0) - f^{\inf}), C_1^2)$. If $\tau = \frac{L_{\max}}{\sqrt{2L}} \sqrt{f(x^0) - f^{\inf}}$, then

$$\min_{k \in [0,K]} \left\| \nabla f(x^k) \right\| \le \sqrt{\frac{2L(f(x^0) - f^{\inf})}{K+1}} + \frac{\sqrt{\left(1 + \frac{C_1\sqrt{2L}}{L_{\max}\sqrt{f(x^0) - f^{\inf}}}\right) C_2}}{(K+1)^{1/2}}$$

$$\le \sqrt{\frac{2L(f(x^0) - f^{\inf})}{K+1}} + \frac{\sqrt{C_2} + \mathcal{O}\left( \max(\sqrt{C_1} \sqrt[4]{f(x^0) - f^{\inf}}, C_1^3/\sqrt{f(x^0) - f^{\inf}}) \right)}{(K+1)^{1/2}}.$$

The first term in the convergence bound of Clip21 matches that of $\alpha$-NormEC as given in (12). However, the second term in the convergence bound of $\alpha$-NormEC is $D/\sqrt{K+1}$, where $D > 0$ can be made arbitrarily small. In contrast, the corresponding term for Clip21 is $C/\sqrt{K+1}$, where $C > 0$ may become significantly larger than $D$ if $x^0 \in \mathbb{R}^d$ is far from the stationary point, leading to a large value of $C_1 = \max_{i \in [1,n]} \left\| \nabla f_i(x^0) \right\|$.

## F. Proof of Theorem 2

To prove Theorem 2, we use Lemma 2, which proves that if $\left\| \nabla f_i(x^k) - g_i^k \right\| \le R$ for some positive scalars $R$, then $\left\| \nabla f_i(x^{k+1}) - g_i^{k+1} \right\| \le R$. Also, we leverage the following lemma, which bounds the difference between the memory vectors maintained by the central server and clients.

**Lemma 3.** *Consider Algorithm 1 for solving Problem (1) in the private setting, where Assumption 1 holds. If $\hat{g}^0 = \frac{1}{n} \sum_{i=1}^n g_i^0$, then*

$$\mathrm{E}\left[ \left\| \hat{g}^{k+1} - \frac{1}{n} \sum_{i=1}^n g^{k+1} \right\| \right] \le \sqrt{\frac{\beta^2(K+1)\sigma_{\mathrm{DP}}^2}{n}}.$$

*Proof.* From the definition of $g^k$ and $\hat{g}^k$,

$$e^{k+1} = e^k + \beta z^{k+1},$$

where $e^k = \hat{g}^k - \frac{1}{n}\sum_{i=1}^n g_i^k$, and $z^k = \frac{1}{n}\sum_{i=1}^n z_i^k$. By applying the equation recursively,

$$e^{k+1} = e^0 + \beta \sum_{l=1}^{k+1} z^l.$$

Therefore, by the triangle inequality,

$$\left\|e^{k+1}\right\| \le \left\|e^0\right\| + \left\|\beta \sum_{l=1}^{k+1} z^l\right\|.$$

If $\hat{g}^0 = \frac{1}{n}\sum_{i=1}^n g_i^0$, then

$$\left\|e^{k+1}\right\| \le \left\|\beta \sum_{l=1}^{k+1} z^l\right\|.$$

Taking the expectation, and using the fact that $\mathrm{E}\left[\langle z^j, z^i\rangle\right] = 0$ for $i < j$ and that $\mathrm{E}\left[\left\|z^k\right\|^2\right] = \frac{\sigma_{\mathrm{DP}}^2}{n}$ ($z_i^k$ is independent of $z_j^k$ for $i \ne j$),

$$
\begin{aligned}
\mathrm{E}\left[\left\|e^{k+1}\right\|\right] &\le \mathrm{E}\left[\left\|\beta \sum_{l=1}^{k+1} z^l\right\|\right] \\
&\le \sqrt{\frac{\beta^2}{n}\sum_{l=1}^{k+1}\sigma_{\mathrm{DP}}^2} \\
&= \sqrt{\frac{\beta^2(k+1)\sigma_{\mathrm{DP}}^2}{n}} \\
&\stackrel{k\le K}{\le} \sqrt{\frac{\beta^2(K+1)\sigma_{\mathrm{DP}}^2}{n}}.
\end{aligned}
$$

$\square$

Now, we prove Theorem 2 in the following steps

**Step 1) Prove by induction that** $\left\|\nabla f_i(x^k) - g_i^k\right\| \le R$ **for** $R = \max_{i\in[1,n]}\left\|\nabla f_i(x^0) - g_i^0\right\|$. For $k = 0$, this is obvious. Next, let $\left\|\nabla f_i(x^l) - g_i^l\right\| \le R$ for $R = \max_{i\in[1,n]}\left\|\nabla f_i(x^0) - g_i^0\right\|$ for $l = 0, 1, \ldots, k$. Then, if $\beta/(\alpha + R) < 1$, and $\gamma \le \frac{\beta R}{\alpha + R}\frac{1}{L_{\max}}$, then from Lemma 2 $\left\|\nabla f_i(x^{k+1}) - g_i^{k+1}\right\| \le R$.

**Step 2) Bound** $\left\|\nabla f_i(x^k) - g_i^{k+1}\right\|$. From the definition of the Euclidean norm,

$$
\begin{aligned}
\left\|\nabla f_i(x^k) - g_i^{k+1}\right\| &\stackrel{g_i^{k+1}}{=} \left\|\nabla f_i(x^k) - g_i^k - \beta\mathrm{Norm}_\alpha\left(\nabla f_i(x^k) - g_i^k\right)\right\| \\
&\stackrel{\text{Lemma 2}}{\le} \left|1 - \frac{\beta}{\alpha + \left\|\nabla f_i(x^k) - g_i^k\right\|}\right|\left\|\nabla f_i(x^k) - g_i^k\right\| \\
&\stackrel{\beta/(\alpha+R)<1}{\le} \left(1 - \frac{\beta}{\alpha + R}\right)R \le R.
\end{aligned}
$$

**Step 3) Derive the descent inequality in** $\mathrm{E}\left[f(x^k) - f^{\inf}\right]$**.** Denote $g^k = \frac{1}{n}\sum_{i=1}^n g_i^k$. By the $L$-smoothness of $f$, and by the definition of $x^{k+1}$,

$$
\begin{aligned}
f(x^{k+1}) - f^{\inf} \quad &\leq \quad f(x^k) - f^{\inf} - \frac{\gamma}{\|\hat{g}^{k+1}\|}\left\langle \nabla f(x^k), \hat{g}^{k+1}\right\rangle + \frac{L\gamma^2}{2} \\[6pt]
&= \quad f(x^k) - f^{\inf} - \gamma\left\|\hat{g}^{k+1}\right\| + \frac{\gamma}{\|\hat{g}^{k+1}\|}\left\langle \nabla f(x^k) - \hat{g}^{k+1}, \hat{g}^{k+1}\right\rangle + \frac{L\gamma^2}{2} \\[6pt]
&\overset{\text{Cauchy-Schwartz inequality}}{\leq} \quad f(x^k) - f^{\inf} - \gamma\left\|\hat{g}^{k+1}\right\| + \gamma\left\|\nabla f(x^k) - \hat{g}^{k+1}\right\| + \frac{L\gamma^2}{2} \\[6pt]
&\overset{\text{triangle inequality}}{\leq} \quad f(x^k) - f^{\inf} - \gamma\left\|\nabla f(x^k)\right\| + 2\gamma\left\|\nabla f(x^k) - \hat{g}^{k+1}\right\| + \frac{L\gamma^2}{2} \\[6pt]
&\overset{\text{triangle inequality}}{\leq} \quad f(x^k) - f^{\inf} - \gamma\left\|\nabla f(x^k)\right\| + 2\gamma\frac{1}{n}\sum_{i=1}^n\left\|\nabla f_i(x^k) - g_i^{k+1}\right\| + 2\gamma\left\|\hat{g}^{k+1} - g^{k+1}\right\| + \frac{L\gamma^2}{2}.
\end{aligned}
$$

Since $\left\|\nabla f_i(x^k) - g_i^{k+1}\right\| \leq R$ with $R = \max_{i\in[1,n]}\left\|\nabla f_i(x^0) - g_i^0\right\|$, we obtain

$$
f(x^{k+1}) - f^{\inf} \leq f(x^k) - f^{\inf} - \gamma\left\|\nabla f(x^k)\right\| + 2\gamma\max_{i\in[1,n]}\left\|\nabla f_i(x^0) - g_i^0\right\| + 2\gamma\left\|\hat{g}^{k+1} - g^{k+1}\right\| + \frac{L\gamma^2}{2}.
$$

Next, by taking the expectation, and by using Lemma 3,

$$
\mathrm{E}\left[f(x^{k+1}) - f^{\inf}\right] \leq \mathrm{E}\left[f(x^k) - f^{\inf}\right] - \gamma\mathrm{E}\left[\left\|\nabla f(x^k)\right\|\right] + 2\gamma\max_{i\in[1,n]}\left\|\nabla f_i(x^0) - g_i^0\right\| + 2\gamma\sqrt{\frac{\beta^2(K+1)\sigma_{\mathrm{DP}}^2}{n}} + \frac{L\gamma^2}{2}.
$$

Therefore,

$$
\begin{aligned}
\min_{k\in[0,K]}\mathrm{E}\left[\left\|\nabla f(x^k)\right\|\right] \quad &\leq \quad \frac{1}{K+1}\sum_{k=0}^K \mathrm{E}\left[\left\|\nabla f(x^k)\right\|\right] \\[6pt]
&\leq \quad \frac{\mathrm{E}\left[f(x^0) - f^{\inf}\right] - \mathrm{E}\left[f(x^{K+1}) - f^{\inf}\right]}{\gamma(K+1)} \\[6pt]
&\qquad + 2\max_{i\in[1,n]}\left\|\nabla f_i(x^0) - g_i^0\right\| + 2\sqrt{\frac{\beta^2(K+1)\sigma_{\mathrm{DP}}^2}{n}} + \frac{L}{2}\gamma \\[6pt]
&\overset{f^{\inf}\geq f(x^{K+1})}{\leq} \quad \frac{f(x^0) - f^{\inf}}{\gamma(K+1)} + 2\max_{i\in[1,n]}\left\|\nabla f_i(x^0) - g_i^0\right\| + 2\sqrt{\frac{\beta^2(K+1)\sigma_{\mathrm{DP}}^2}{n}} + \frac{L}{2}\gamma.
\end{aligned}
$$

# G. Proof of Corollary 2

Let $\sigma_{\mathrm{DP}} = \mathcal{O}\left(\frac{\sqrt{(K+1)\log(1/\delta)}}{\epsilon}\right)$. Then, if we choose $\beta = \frac{\beta_0}{K+1}$ with $0 < \beta_0 < \alpha + R$, then $\gamma \leq \frac{\beta_0 R}{\alpha+R}\frac{1}{L_{\max}}\frac{1}{K+1}$ with $R = \max_{i\in[1,n]}\left\|\nabla f_i(x^0) - g_i^0\right\|$, and

$$
\min_{k\in[0,K]}\mathrm{E}\left[\left\|\nabla f(x^k)\right\|\right] \leq \frac{L_{\max}(\alpha+R)(f(x^0) - f^{\inf})}{\beta_0 R} + 2R + \mathcal{O}\left(\frac{\beta_0\sqrt{\log(1/\delta)}}{\sqrt{n}\epsilon}\right) + \frac{L\beta_0 R}{2(\alpha+R)L_{\max}}\frac{1}{K+1}.
$$

In addition, if $\beta_0 \leq \sqrt{\frac{L_{\max}(\alpha+R)(f(x^0) - f^{\inf})}{R}}\frac{\sqrt[4]{n}\sqrt{\epsilon}}{\sqrt[4]{d}\sqrt[4]{\log(1/\delta)}}$, and $\alpha > \beta_0$, then

$$
\min_{k\in[0,K]}\mathrm{E}\left[\left\|\nabla f(x^k)\right\|\right] \leq 2R + \mathcal{O}\left(\sqrt{\frac{L_{\max}(\alpha+R)(f(x^0) - f^{\inf})}{R}}\frac{\sqrt[4]{d}\sqrt[4]{\log(1/\delta)}}{\sqrt[4]{n}\sqrt{\epsilon}}\right) + \mathcal{O}\left(\frac{1}{K+1}\right).
$$

## H. Experimental details and additional results

**Additional details.** All the methods are run with constant step size (learning rate) without the use of techniques like schedulers, warm-up, or weight decay The dataset is split into train (90%) and test (10%) parts. The train samples are randomly shuffled and distributed across 10 workers. Every worker computes gradients with batch size 32. The training is performed for 300 communication rounds. The random seed was fixed to 42 for reproducibility.

**Hyper-parameters selection.** We evaluate the following combinations of hyper-parameters:

- step size $\gamma$: $\{0.001, 0.01, 0.1, 1.0\}$,

- Sensitivity/clip threshold $\beta$: $\{0.01, 0.1, 1.0, 10.0\}$,

- $\alpha$ values: $\{0.01, 0.1, 1.0\}$.

Our implementation is based on the public GitHub repository of Idelbayev. Experiments were performed on a machine with single GPU: NVIDIA GeForce RTX 3090.

**H.1. Sensitivity of $\alpha$-NormEC to parameters $\beta, \alpha$**

Similarly to Figure 1 (with Accuracy) minimal training loss is displayed in Figure 4. We also show final metrics (at the end of training) in Figure 5 (Accuracy) and in Figure 6 (Loss). These additional plots are consistent with result in Figure 1.

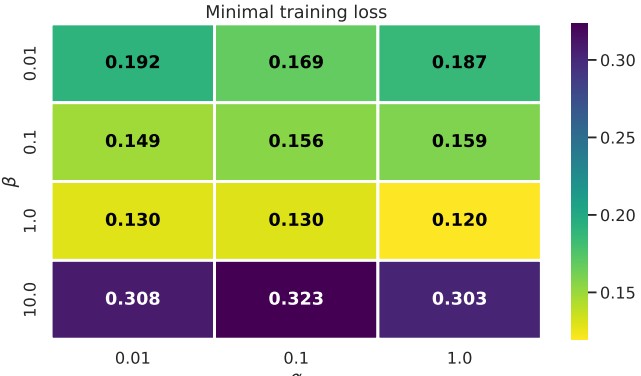

Figure 4: **Minimal** train loss achieved $\alpha$-NormEC.

Figure 7 shows convergence curves which confirm our prior observations that choice of $\alpha$ has a small effect on the method's performance as the variations for each $\beta$ are minor. Especially for the test accuracy results. Interestingly, some of the convergence curves intersect, which means that the optimal set of parameters may depend on the stopping time of the method. Namely, $\beta = 0.1$ results in the fastest convergence until epoch 170 but later is overtaken by $\beta = 1$. A similar picture is observed for a pair of curves at $\beta = 10$ and $\beta = 0.01$ but for smaller number of communication rounds $k \sim 50$.

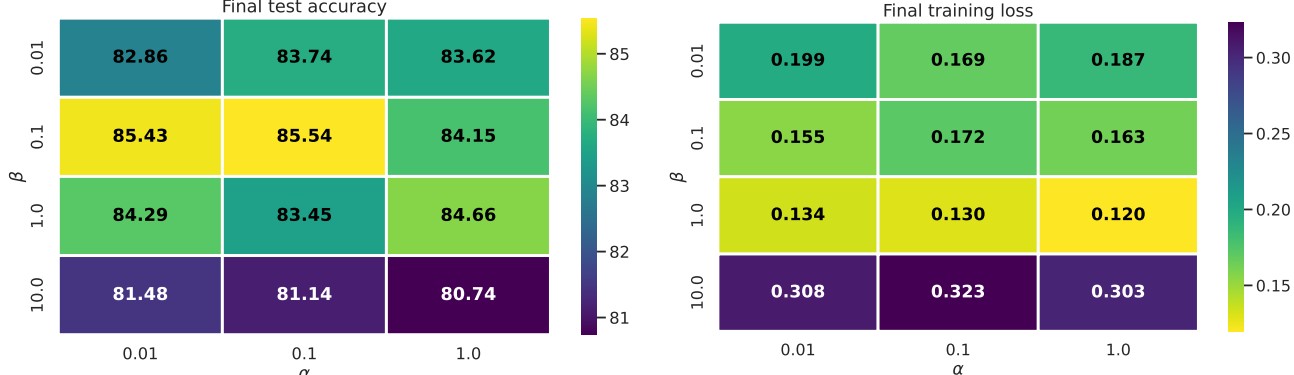

Figure 5: **Final** test accuracy achieved $\alpha$-NormEC.

Figure 6: **Final** train loss achieved $\alpha$-NormEC.

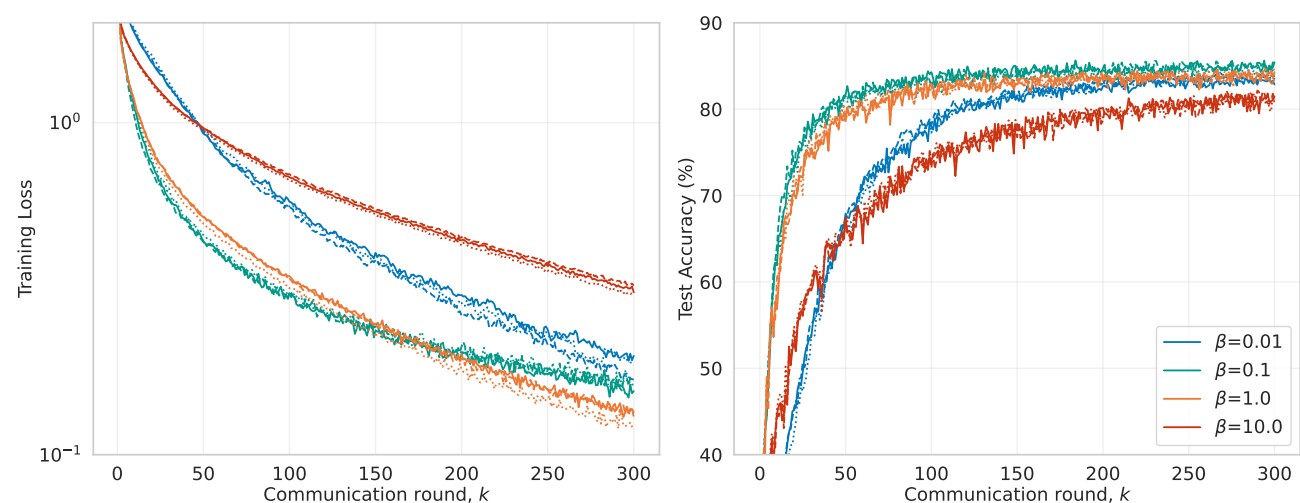

Figure 7: $\alpha$-NormEC convergence for varying parameters $\beta$ and $\alpha$. For each $\beta$ value, solid lines correspond to $\alpha = 0.01$, dashed lines to $\alpha = 0.1$, and dotted lines to $\alpha = 1.0$.

## H.2. Benefits of Error Compensation

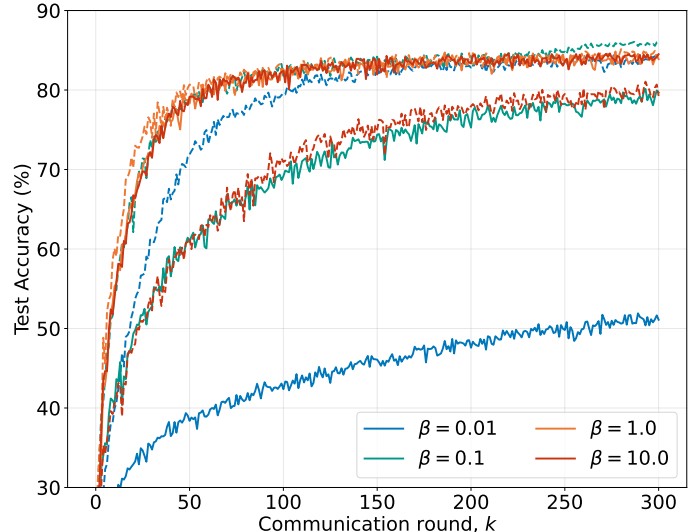

Figure 8: Comparison of DP-SGD (2) [solid] and $\alpha$-NormEC (1) [dashed] without server normalization.

| Method | $\beta$ | $\gamma$ | Final Accuracy |
|---|---|---|---|
| $\alpha$-NormEC | 0.01 | 0.1 | 84.04% |
| | 0.1 | 0.1 | **86.09**% |
| | 1.0 | 0.1 | 84.80% |
| | 10.0 | 0.01 | 79.25% |
| DP-SGD (2) | 0.01 | 1.0 | 51.10% |
| | 0.1 | 1.0 | 79.68% |
| | 1.0 | 1.0 | 83.89% |
| | 10.0 | 0.1 | 84.50% |

Figure 9: Best configurations and final test accuracies.

The test accuracy curves in Figure 8 reveal that Error Compensation (EC) not only improves convergence speed but also leads to better final performance. This is particularly evident for small $\beta$ values ($\beta = 0.01$), where DP-SGD achieves only 51.10% accuracy while $\alpha$-NormEC reaches 84.04%. Table 9 shows that $\alpha$-NormEC consistently outperforms DP-SGD across most configurations, achieving the best accuracy of 86.09% at $\beta = 0.1$. The only exception is at $\beta = 10.0$, though this setting is less practical due to privacy considerations.

These comprehensive results demonstrate that EC provides substantial improvements in both optimization dynamics and final model quality, while maintaining robustness across different parameter settings.

## H.3. Effect of server normalization

We conduct an ablation study to analyze the impact of server-side normalization (Line 11 in Algorithm 1) on $\alpha$-NormEC performance. Figure 10 illustrates the convergence behavior through training loss and test accuracy curves, while Table 2 summarizes the optimal hyper-parameters and final accuracies.

Our analysis reveals that server normalization has a more nuanced effect on performance compared to Error Compensation. The impact varies across different $\beta$ values:

- For large $\beta = 10.0$, server normalization proves beneficial, improving accuracy by approximately 2.2.

- For moderate to small $\beta$ values ($\beta \in \{0.01, 1.0\}$), omitting server normalization yields slightly better results.

- Most notably, at $\beta = 0.1$, the method without server normalization achieves optimal performance of **86.09%**.

These results suggest that while server normalization can be helpful in certain regimes (particularly with large $\beta$), it is not universally beneficial. The choice of whether to employ server normalization should be guided by the selected $\beta$ value, with smaller $\beta$ values generally performing better without this additional normalization step.

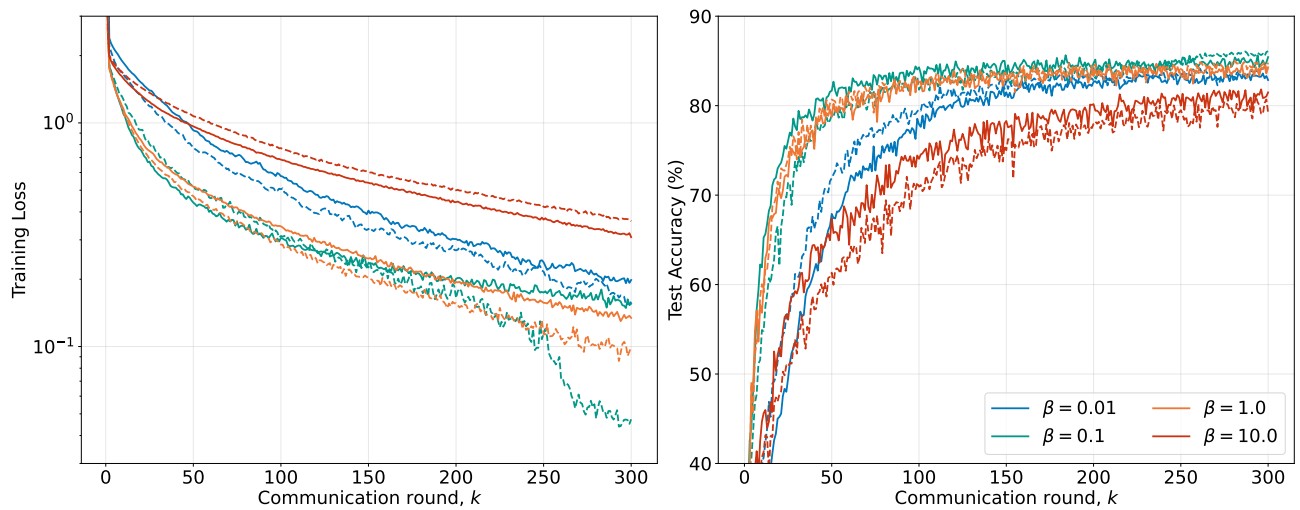

Figure 10: $\alpha$-NormEC with [solid] and without [dashed] server normalization.

| Method: $\alpha$-NormEC | $\beta$ | $\gamma$ | Final Accuracy |
|---|---|---|---|
| With server normalization | 0.01 | 0.01 | 82.86% |
| | 0.1 | 0.1 | 85.43% |
| | 1.0 | 0.1 | 84.29% |
| | 10.0 | 0.1 | 81.48% |
| Without server normalization | 0.01 | 0.1 | 84.04% |
| | 0.1 | 0.1 | **86.09**% |
| | 1.0 | 0.1 | 84.80% |
| | 10.0 | 0.01 | 79.25% |

Table 2: Best configurations and final test accuracies.

## H.4. Comparison of Clip21 and $\alpha$-NormEC

The experimental results, shown in Figure 11, demonstrate that both methods achieve comparable performance across most $\beta$ values. For moderate values of $\beta$ (0.1 and 1.0), both methods show similar convergence patterns and final accuracies, with $\alpha$-NormEC achieving marginally better results (**86.09%** vs 85.91% at $\beta = 0.1$).

The methods show different behaviors at extreme $\beta$ values. At small $\beta = 0.01$, $\alpha$-NormEC demonstrates better performance (84.04% vs 83.00%), suggesting more stable training under aggressive normalization. Conversely, at large $\beta = 10.0$, Clip21 maintains better performance (83.19% vs 79.25%), probably because the clipping is so large that it almost never happens.

Both methods achieve their best performance with $\gamma = 0.1$ in most cases, except for $\alpha$-NormEC at $\beta = 10.0$ where a smaller learning rate ($\gamma = 0.01$) was optimal. Note that we run $\alpha$-NormEC without server normalization is it showed better performance according to Appendix H.3.

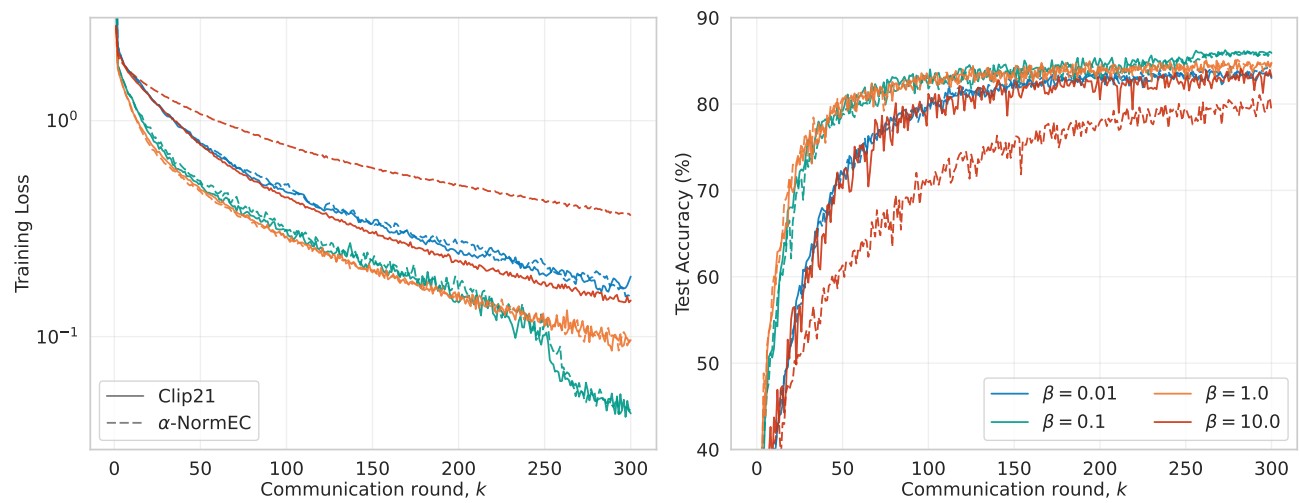

Figure 11: Comparison of Clip21 [solid] and $\alpha$-NormEC [dashed].

| Method | $\beta$ | $\gamma$ | Final Accuracy |
|---|---|---|---|
| Clip21 | 0.01 | 0.1 | 83.00% |
| | 0.1 | 0.1 | 85.91% |
| | 1.0 | 0.1 | 84.78% |
| | 10.0 | 0.1 | 83.19% |
| $\alpha$-NormEC | 0.01 | 0.1 | 84.04% |
| | 0.1 | 0.1 | **86.09%** |
| | 1.0 | 0.1 | 84.80% |
| | 10.0 | 0.01 | 79.25% |

Table 3: Best configurations and final test accuracies for Clip21 and $\alpha$-NormEC methods.

## H.5. Differentially Private results

The training loss trajectories in Figure 12 provide further evidence that smaller $\beta$ values enable more effective optimization under privacy constraints, with $\beta = 0.01$ achieving the fastest convergence and lowest final loss values.

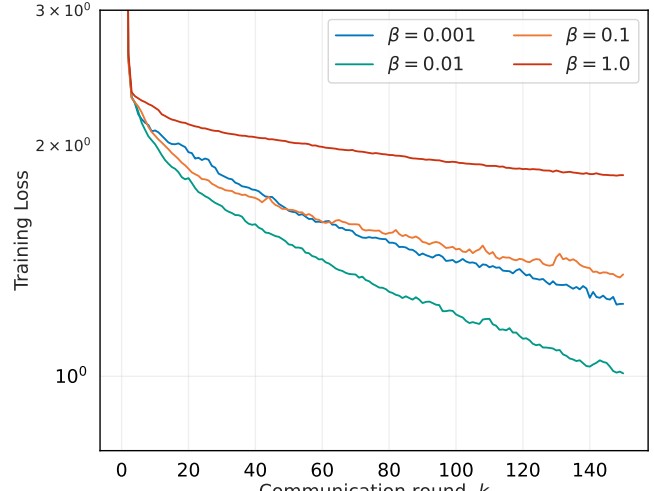

Figure 12: Convergence of DP-$\alpha$-NormEC.

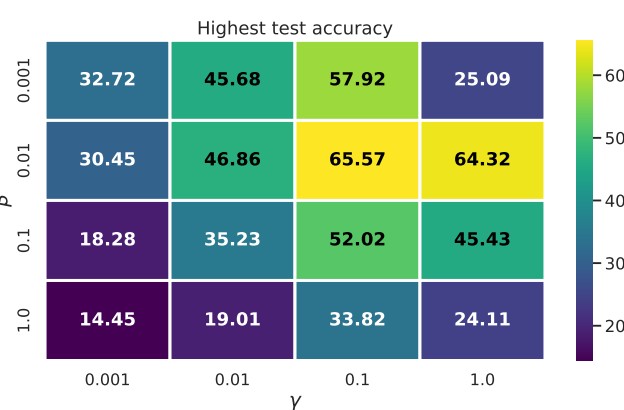

Figure 13: Best configurations and highest test accuracies.

