# OpenReview forum: "Smoothed Normalization for Efficient Distributed Private Optimization"
_ICML.cc/2025/Conference — Submitted to ICML 2025_

### Official Review · Reviewer_LMxv · 2025-02-15

**Overall Recommendation:** 4

**Summary:**

This work focuses on differential privacy in the federated learning. It mentions clipping-based DP-FL optimization like DP-SGD is hard to converge due to clipping bias, especially for non-convex, smooth problems. Instead of (adaptive) clipping, this work chooses smoothed normalization to tackle the problem by proposing a method called alpha-NormEC, which combines smoothed normalization and error feedback.

**Claims And Evidence:**

This work walks through problem formulation and one de-facto popular solution, DP-SGD, in a good way. By discussing limitations (e.g., less convergence) of related works, this work points out clipping approach does not quite consider clipping bias or introduce too many constraints, especially in the federated learning. The necessity of this work is well pronounced based on success of smoothed normalization in single-node DP setting and good-but-can-be-better error feedback methods. Such combination can lead a good result to tackle the convergence of non-convex and smooth functions.

**Essential References Not Discussed:**

In the beginning of the paper, it said “no DP distributed method for smooth non-convex optimization problems”. However, to my knowledge, "DIFF2” paper cited in this work (published on ICML 2023) has discussed the problem for federated learning and provide a feasible framework. Can you explain more on the difference between DIFF2 paper and yours?

**Experimental Designs Or Analyses:**

Although image classification is a good task to conduct experiments, the CIFAR-10 dataset is pretty small and quite outdated in 2025. Also considering current LLM trending, ResNet20 is also a less convincing architecture to work on. The scalability is a very important aspect to make the work more realistic and useful than the prototype level.

**Methods And Evaluation Criteria:**

The methods in this work is more like an incremental approach since it is highly relied on the existing mechanism like EF21. But the idea to introduce smoothed normalization is a great to make the mechanism converge faster even with bias in the distributed (federated) learning. The evaluation on image classification task is a good criteria to measure performance of DP-related problems.

**Other Comments Or Suggestions:**

1. In the preliminaries, 3.3 discussed too much details in DP-SGG. The audience of this paper could DP-related researcher on ML, so DP-SGD probably is a default knowledge. I suggest to move details of DP-SGD in the appendix and mainly focuses on limitations.
2. As discussed in the experimental design section, decode-only (e.g., transformer) models may be better than ResNet20 to produce convincing results to LLM-era researchers.

**Other Strengths And Weaknesses:**

Strengths:

1. Hyperparameters to tune the alpha-NormEC are easier to implement and faster to convergence for non-convex and smooth functions than prior works. And it requires no additional restrictions.
2. DP version of alpha-NormEC guarantees convergence even considering DP-related bias.

Weakness:

1. Besides dataset and model architecture, this work put too much experimental details to the appendix. For example, comparison to existing work like Clip21 is a good evidence to convince audience on the effectiveness of this work instead of only reading theoretical parts.
2. This work just discusses the difference with pure clipping approach in the text rather than conducting any comparison experiments (not even mentioned in the appendix).

**Questions For Authors:**

N/A

**Relation To Broader Scientific Literature:**

1. This work points out neglect of clipping bias in the current DP-SGD-related works, which are important for private optimization (not just FL) to converge.
2. Although smoothed normalization is a promising approach to tackle the issue, the cited work Yang et al. (2022) looks plausible and is rejected by TMLR last year, which contains fatal errors (For more info, take a look this link https://openreview.net/forum?id=wLg9JrwFvL). Probably that paper is not a good work as a main citation in the introduction.
3. Bu et al. (2024) is a great work to cite and is highly related to this work. That work provides convinced results on smoothed normalization for non-federated case. However, the assumptions are restricter than this work and is not trivial to extend to federated learning.
4. This work also well studied advantages and limitations of distributed/single-node non-private methods related to clipping.
5. Error feedback as a main course in this work is also well-surveyed.

**Theoretical Claims:**

I carefully walked through the Theorems and proofs. To my best knowledge, those claims are convinced to me.

---

> ### Author Rebuttal · Authors · 2025-04-01
>
> Dear reviewer LMxv,
>
> We appreciate your time, effort, and thoughtful feedback.
>
> We thank you for your appreciation towards the contributions of our work on leveraging smoothed normalization and error feedback to design distributed algorithms with the first provable convergence under the privacy budget.
>
> > In the beginning of the paper, it said “no DP distributed method for smooth non-convex optimization problems”. However, to my knowledge, "DIFF2” paper cited in this work (published on ICML 2023) has discussed the problem for federated learning and provide a feasible framework. Can you explain more on the difference between DIFF2 paper and yours?
>
>
> DIFF2 assumes both smoothness of objective functions and bounded gradient conditions to derive the convergence. Unlike DIFF2, DP-$\alpha$-NormEC achieves the utility guarantee under smoothness without bounded gradient conditions.
>
> Next, on the one hand, DIFF2 uses local gradient differences $\nabla f_i(x^{k}) - \nabla f_i(x^{k-1})$, and, thus, requires computing two gradients at different points at every iteration. On the other hand, $\alpha$-NormEC privatizes the difference between the local gradient and memory vector $\nabla f_i(x^{k}) - g_i^{k}$. Moreover, while DIFF2 adds the private noise on the server, $\alpha$-NormEC adds the noise to the updates from the clients before being communicated to the server.
>
> > Although smoothed normalization is a promising approach to tackle the issue, the cited work Yang et al. (2022) looks plausible and is rejected by TMLR last year, which contains fatal errors (For more info, take a look this link https://openreview.net/forum?id=wLg9JrwFvL). Probably that paper is not a good work as a main citation in the introduction.
>
> Thank you for pointing us out. We will remove Yang et al. (2022) as a main citation in the introduction section, but keep it in the related work section.
>
>
> > Besides dataset and model architecture, this work put too much experimental details to the appendix. For example, comparison to existing work like Clip21 is a good evidence to convince audience on the effectiveness of this work instead of only reading theoretical parts.
>
> We agree with this suggestion. We will move results in the ablation study from the appendix into the main text, such as the results showing that $\alpha$-NormEC provides slightly strong convergence performance than Clip21 in both non-private and private settings.
>
> > This work just discusses the difference with pure clipping approach in the text rather than conducting any comparison experiments (not even mentioned in the appendix).
>
> In Appendix H.2, we compare DP-SGD and $\alpha$-NormEC. Our results indicate that Error Compensation significantly improves over the baseline.
>
> > In the preliminaries, 3.3 discussed too much details in DP-SGG. The audience of this paper could DP-related researcher on ML, so DP-SGD probably is a default knowledge. I suggest to move details of DP-SGD in the appendix and mainly focuses on limitations.
>
>
> Thank you for your suggestion. We will improve Section 3.3 by moving DP-SGD details to the appendix, allowing for a more detailed explanation of its convergence limitations. This revision will be included in the next version.
>
>
> > As discussed in the experimental design section, decode-only (e.g., transformer) models may be better than ResNet20 to produce convincing results to LLM-era researchers.
>
> We appreciate your comment. We agree that DP-$\alpha$-NormEC will be of huge interest in private training of LLM models. We will provide additional experiments on training transformer models in the revised manuscript.
>
>
>
> We hope these clarifications have sufficiently addressed your concerns, providing a clearer understanding of our study's contributions and implications. We are eager to engage in further discussion to resolve any remaining concerns. Please consider the score accordingly.
>
> Best Regard,
>
> Authors

---

### Official Review · Reviewer_yzkj · 2025-03-10

**Overall Recommendation:** 2

**Summary:**

This paper proposes a distributed optimization algorithm (called α-NormEC). It uses smoothed normalization with error feedback to solve non-convex, smooth optimization problems in both non-private and differentially private settings. The method claims to achieve provable convergence guarantees without requiring bounded gradient norms. It claims to be the first method to provide convergence guarantees for private training under standard assumptions, addressing the challenges of clipping bias in distributed differentially private optimization. The paper also demonstrates empirical performance on practical tasks.

**Claims And Evidence:**

Overall, the paper is fine (the problem is valid, and the solution seems to be working, although I cannot test it directly or replicate it), but there are certain elements that seem to be oversimplified or glossed over, which, if not the case, then are poorly explained.

The proposed algo provides convergence guarantees (theoretically), but the baseline assumption is the objective functions' smoothness and bounded from below. But what if these are not in real-world scenarios, like heterogeneous data noisy gradients to begin with, or non-smooth func as in some DL? The paper could have benefited from including a deeper analysis and providing better arguments in this regard.

The paper makes a point of highlighting the issues with the gradient clipping technique. But, there is no mention of adaptive or data-driven clipping techniques. Once again, an analysis or comparison of smoothed normalization and the adaptive clipping methods would have clarified this. It would have actually shown if the proposed work is even better than or at par with them.

**Essential References Not Discussed:**

Note, that it is not sufficient that the paper cites another work, but also that it discusses the the proper context.
Andrew et al., "Differentially Private Learning with Adaptive Clipping," NeurIPS 2019: cited, but has to be compared to.
AdaCliP: Adaptive Clipping for Private SGD (Pichapati et al., 2019)
Gradient Sparsification for Efficient Wireless Federated Learning with DP (Wei et al., 2023)
Adaptive Gradient Sparsification for Efficient FL (Han et al., 2020)

**Experimental Designs Or Analyses:**

The dataset diversity is limited, and hence the question of generalizability arises. Comparisons with other better baselines could also have improved and highlighted the work's impact. Real-world applications will require communication and other computational overheads, hence the feasibility is still a question (although I understand this may not be a huge issue). A more elaborate privacy-utility analysis could be beneficial, especially for extreme privacy settings (financial, medical, etc.)

**Methods And Evaluation Criteria:**

Expanding on the same point of verboseness of the paper, the paper could have benefited from better evaluations.
Currently, it is very specific to CIFAR-10  using ResNet20. This feels like just proof of concept type work.

The paper positions itself that the application of this work is in federated learning; however, it does not add the scalability of communication efficiency. The argument can be that the paper focuses on providing the non-convex solution more as compared to scalability and efficiency; however, then it can be countered that this work is not only for federated learning and can be applied to other domains, hence its relevance to ICML is week. NLP, time series forecasting, reinforcement learning, or something similar could have been used to demonstrate its application better.

The paper should provide a more detailed analysis or experiments showing how the method performs as ϵ approaches higher values (e.g., closer to 1 or higher) and the impact of this on both privacy and utility. The paper lacks a clear discussion of the trade-off between privacy loss and model performance

**Other Comments Or Suggestions:**

A symbol table would have been beneficial, given the extensive appendices.

Typo P1: To enfore DP

**Other Strengths And Weaknesses:**

Strengths:
The paper provides the convergence guarantee for private training without assuming bounded gradient norms.
Presents a rigorous mathematical treatment of α-NormEC and its convergence properties.
Introduction of smoothed normalization as a substitute for gradient clipping is an interesting idea.
α-NormEC can be applied in distributed settings, making it relevant to federated learning.

Weakness:
Paper does not benchmark against recent advances in adaptive clipping and gradient sparsification.
Does not fully analyze how privacy noise affects training under different privacy budgets (ϵ, δ).
Weak empirical evaluation.
No Discussion of Momentum and Adaptive Learning Rates in DP-SGD.

**Questions For Authors:**

1- α-NormEC performance at low privacy budgets?
2- How does α-NormEC compare to adaptive clipping?
3- Elaborate on computational and communication costs.
4- Gradient sparsification as a comparison, and α-NormEC applied to non-iid data in FL?

**Relation To Broader Scientific Literature:**

For the most part, I am satisfied by the literature review, although it is quite lengthy. However, as mentioned earlier, the adaptive clipping literature and its comparison would be beneficial.

**Theoretical Claims:**

Not all of them. The proofs are primarily listed in the appendices (which are longer than the paper itself). The proofs assume smoothness and bounded loss functions, which may not always hold in practical (real-world) non-convex optimization problems.

---

> ### Author Rebuttal · Authors · 2025-04-01
>
> Dear reviewer yzkj,
>
> We appreciate your time, effort, and thoughtful feedback.
>
> > The proposed algo provides convergence guarantees (theoretically), but the baseline assumption is the objective functions' smoothness and bounded from below.
>
> Our  assumptions are standard for analyzing distributed algorithms on non-convex problems, e.g. by Nesterov et al. (2018) and [A]. Furthermore, in the context of differential privacy, we do not impose bounded gradients, which are restrictive but used by prior literature.
>
> [A] Karimireddy, S. P., Kale, S., Mohri, M., Reddi, S., Stich, S., & Suresh, A. T. (2020, November). Scaffold: Stochastic controlled averaging for federated learning. In International conference on machine learning (pp. 5132-5143). PMLR.
>
>
> ​​>Weak empirical evaluation.
>
> Although our main focus is on algorithmic development, our work contains an extensive experimental section that evaluates several algorithms in different settings, including the ablation studies of the design components of our method. Furthermore, our experimental setting is standard in the literature on differential privacy (Papernot et al., 2020; Li and Chi, 2023; Allouah et al., 2024). Can you elaborate? We will be happy to incorporate your constructive suggestions.
>
>
> > 1- α-NormEC performance at low privacy budgets?
>
> We ran $\alpha$-NormEC across a range of $\beta$ values in Figure 3, which shows a privacy-utility trade-off. Lower $\beta$ results in smaller added noise, resulting in higher privacy loss and decreased performance of the model. The effect of tuning $\beta$ was observed also for the non-private version of $\alpha$-NormEC (without DP noise). Furthermore, we performed additional comparisons of the methods for  the private training, where DP-$\alpha$-NormEC significantly outperforms DP extension of Clip21.  Kindly see  [the attached plot link](https://postimg.cc/bDhgkq7R).
>
>
> > 2- How does α-NormEC compare to adaptive clipping?
> > … there is no mention of adaptive or data-driven clipping techniques.
>
>
> We develop distributed algorithms with the first provable convergence without relying on restrictive bounded gradient assumptions. Adaptive clipping is orthogonal to our work, due to smoothed normalization that removes the need to tune the private noise.
>
> In Paragraph 3 of introduction,  we reviewed adaptive clipping techniques, such as by Andrew et al., (2019), and recent results by Merad & Gaïffas (2023) and Shulgin & Richtárik (2024).  Andrew et al., (2019) does not provide any guarantees we can compare to. Shulgin & Richtárik (2024) showed the first theoretical analysis only in the centralized (single node) case, and showed that SGD with Adaptive clipping suffers from a similar non-convergence issue as SGD with constant clipping (Koloskova et al., 2023). Moreover, Pichapati et al., (2019) considers coordinate-wise adaptive clipping for AdaCliP that is orthogonal to our work, and presents  its analysis of the single-node case under the bounded gradient assumption, the condition our distributed algorithms do not require
>
> >3- Elaborate on computational and communication costs.
>
> At every iteration, each client computes the local gradient, updates local memory $g_i^{k+1}$, and sends the privatized difference between the gradient and memory vector to the server. Then, the server performs averaging and sends the updated model back to the clients. There is no additional communication or computational overhead compared to distributed SGD.
>
> >4- Gradient sparsification as a comparison, and α-NormEC applied to non-iid data in FL?
>
> Compression (Sparsification) techniques are complementary to our work, as compression does not preserve privacy. Sparsification does not guarantee bounded sensitivity and has to be combined with clipping or normalization for differential privacy. We believe $\alpha$-NormEC can be combined with compression to further improve communication efficiency while it preserves privacy. $\alpha$-NormEC with compression can be used to train over arbitrarily heterogeneous and non-iid data, thanks to error compensation mechanisms similar to EF21 that remove the uniformly bounded heterogeneity assumptions.
>
>
>
> > No Discussion of Momentum and Adaptive Learning Rates in DP-SGD.
>
> We will add the discussion on techniques like momentum and step size DP-SGD in the revision as they can be beneficial for empirical performance. Error Compensation can be viewed as a variation of server momentum equivalent to classical heavy-ball momentum up to reparametrization [B] .
>
> [B] Garrigos, Guillaume, and Robert M. Gower. "Handbook of convergence theorems for (stochastic) gradient methods." arXiv preprint arXiv:2301.11235 (2023).
>
> Next, thank you for the typos and suggestions. We will reorganize the initial part of the paper and shorten the literature review to include more results in the main part.
>
> We hope these clarifications have addressed your concerns. Please consider the score accordingly.
>
> Best Regard,
>
> Authors

---

### Official Review · Reviewer_BGLy · 2025-03-12

**Overall Recommendation:** 3

**Summary:**

This paper studies federated learning with gradient clipping in the non-private and private settings. In the non-private setting, their algorithm matches existing results for clipped methods. In the private setting, to my knowledge, their convergence results are new.

**Claims And Evidence:**

Their authors claim their method matches existing theoretical convergence guarantees and empirically is more practical. They provide proofs for the theoretical claims and experiments to back up the empirical performance of the algorithm. One argument they make against prior clipped SGD methods is high sensitivity to the clipping norm, whereas their method is less sensitive to the (new) hyper-parameters. They provide experiments to justify this.

**Essential References Not Discussed:**

I think the literature on non-federated non-convex optimization could be discussed more. For example, the idea of using gradient differences in smooth non-convex optimization goes back to at least [1]. I understand there is nuance in the distributed setting, but it seems to me this nuance is lost since each client is computing its gradient at the same local iterate, x^k, each round.

[1]: Spider: Near-Optimal Non-Convex Optimization via Stochastic
Path Integrated Differential Estimator

**Experimental Designs Or Analyses:**

The design of the experiments which are included makes sense.

**Methods And Evaluation Criteria:**

The method is evaluated through a formal convergence rate and experiments.

**Other Comments Or Suggestions:**

At line 267, the authors remark that the Clip21 algorithm from Clip21 algorithm achieves 1/K, I assume they mean 1/sqrt{K} as per the discussion in the Appendix?

**Other Strengths And Weaknesses:**

The presentation of the paper is very nice, and makes for easy reading. The authors also do a great job describing clearly the shortcomings of previous work and their contributions.

I think the experiments do a good job of addressing obvious follow up questions that would arise for proposing a method like this.

With that said, the spirit of DP SGD with clipping is to avoid assuming regularity conditions on the loss. It's true that the authors avoid Lipschitzness, but they do so only at the expense of imposing smoothness, which is arguably even harder to come by in modern methods.

**Questions For Authors:**

How is the initial gradient being made private? It is strange that it is passed in as a parameter to the algorithm.

Relatedly, I have some concern with the premise of the paper. The authors perform all their updates using gradient differences, and also assume the loss is smooth. In the DP setting, why would we ever need to clip if this were the case? The sensitivity of the updates is already bounded by smoothness. For example, one could do something like Algorithm 1 in [1] (but without "resetting" the gradient estimator).

Further, the authors claim their result is the first convergence rate for DP distributed learning. But their algorithm involves communicating x^k to each client every round. At that point, it seems DP federated learning is a more fair comparison, for which there are many existing convergence results.

[1]: Faster Rates of Convergence to Stationary Points in
Differentially Private Optimization

**Relation To Broader Scientific Literature:**

This works builds on previous work that has studied clipping in both federated/non-federated and private/non-private settings. In particular, the method the authors propose seems to be a careful combination of previous techniques, error feedback and smoothed normalization.

To be honest, this feels more like a federated learning paper than a distributed learning paper. At every iteration, each client is computing a gradient at the same globally known point, x^k. If the communication to do this is happening, why would existing federated methods not readily apply?

**Theoretical Claims:**

The privacy and convergence proofs look correct from my reading.

---

> ### Author Rebuttal · Authors · 2025-04-01
>
> Dear reviewer BGLy,
>
> We appreciate your time, effort, and thoughtful feedback.
>
> > At every iteration, each client is computing a gradient at the same globally known point, x^k.
>
> Could you please specify the federated methods? Because federated learning algorithms exchange the local model updates, not the gradients as required by $\alpha$-NormEC. However, $\alpha$-NormEC can be modified to FedAvg algorithms. The generalization of our methods to federated settings presents a challenging direction, as it requires modifications of our current analysis.
>
>
> > At line 267, the authors remark that the Clip21 algorithm from Clip21 algorithm achieves 1/K, I assume they mean 1/sqrt{K} as per the discussion in the Appendix?
>
> We apologize for this misunderstanding. At line 267, Clip21 attains the $\mathcal{O}(1/K)$ convergence of the **squared** gradient norm $\|| \nabla f(x) \||^2$. This statement is equivalent to the $\mathcal{O}(1/\sqrt{K})$ convergence in the gradient norm of Clip21 in Appendix E.
>
> We will revise the discussion at Line 267.
>
>
> > How is the initial gradient being made private?
>
> **Privatization of the difference between the local gradient and memory:** As the initial gradient $\nabla f_i(x^0)$ is not shared with the server, it is not needed to be privatized. Only the **difference** between local gradient and memory vector at the initialization $\nabla f_i(x^0) - g_i^0$ is privatized and sent to the server with the gaussian mechanism.
>
> **Initialization of the memory vectors:** The memory vector $\hat{g}^0$ on the server is initialized as $\frac{1}{n} \sum_{i=1}^n g_i^0$ (Line 948). Our analysis from Lemma 3 allows easy extension to arbitrary initialization,e.g. $\hat{g}^0 = \frac{1}{n}\sum_{i=1}^n g_i^0 + e$. Here, this additional error term $e$ can be small if we privately estimate the mean of vectors $g_i^0$, which  incur once and thus a small privacy loss (compared to the iterative process). Furthermore, secure aggregation techniques can eliminate this error entirely. For instance, if clients agree on a shared random seed, they can add and subtract cryptographic noise to their local memory vectors, respectively, without affecting the average. Consider two clients with local memory vectors $g_1^0$ and $g_2^0$.  The first client can add cryptographic noise $h$ to $g_1^0$ and the second client subtract $h$ from $g_2^0$. This would protect the vectors $g_i^0$ from the server but the average would be exactly the average as $\frac{1}{2}(g_1^0 + h) + \frac{1}{2}(g_2^0 - h) = \frac{1}{2}(g_1^0 + g_2^0)$.
>
>
>
>
> > Relatedly, I have some concern with the premise of the paper. The authors perform all their updates using gradient differences, and also assume the loss is smooth. In the DP setting, why would we ever need to clip if this were the case? The sensitivity of the updates is already bounded by smoothness.
>
>  We assume smoothness of the loss function $\||\nabla f(x)-\nabla f(y)\|| \leq L\||x-y\||$, the most standard assumption in non-convex optimization literature. It does not imply bounded sensitivity of the gradient $\||\nabla f(x)\||$ unless the domain is bounded ($\||x-y\|| \leq \mathcal{D}$) which is a restrictive condition. Existing literature considers the convergence of DP distributed algorithms for minimizing smooth losses by further imposing bounded gradient conditions. This restricts the class of optimization problems. These bounded gradients can be achieved by assuming either Lipschitz continuous functions (as in [1]) or bounded domain. These conditions allow us to bound the sensitivity of the updates without applying clipping or normalization. However, this sensitivity is impossible to compute for many loss functions used in training machine learning models. Even when it can be estimated, its bound is often overly pessimistic, thus leading to excessively large DP noise and thus significantly degrading the algorithmic convergence performance.”
>
> > Further, the authors claim their result is the first convergence rate for DP distributed learning. But their algorithm involves communicating x^k to each client every round.
>
> We believe there is a misunderstanding. Our algorithms communicate the normalized gradient difference $N_{\alpha}(\nabla f_i(x^k)-g_i^k)$, not the iterates $x^k$. Furthermore, $\alpha$-NormEC attains convergence under privacy budget for minimizing smooth functions without assuming bounded gradients and/or ignoring the effect of normalization, which existing literature often requires.
>
> To our knowledge, only Das et al. (2022) and Li et al. (2024) analyze DP federated methods without restrictive bounded gradient assumption. However, these algorithms (Algorithm 1 of Das et al. (2022)) with one local step (E=1) result in DP distributed clipped gradient methods. These algorithms using clipping/normalization even without the DP noise do not converge for simple examples (see Section 3.4). To fix the convergence issue, we leverage error feedback.
>
>
> Best regards,
>
> Authors

---

> > ### Comment · Reviewer_BGLy · 2025-04-07
> >
> > Thank you for the detailed response. To follow up on the point about the gradient updates, my point was that you don't need to bound the norm of the individual gradients to bound the sensitivity of the gradient difference estimate. If you estimated gradient differences using $\nabla f_i(x^k) - \nabla f_i(x^{k-1})$, the sensitivity of this vector is bounded via smoothness. Is there a good reason not to do this in the setting you consider?

---

> > > ### Author Response · Authors · 2025-04-08
> > >
> > > Dear Reviewer BGLy,
> > >
> > > Thank you for the follow-up question. We appreciate the opportunity to clarify the design choices in `𝛼-NormEC`.
> > >
> > > In `𝛼-NormEC`, each client does **not** compute the gradient difference $\nabla f_i(x^k)-\nabla f_i(x^{k-1})$. Instead, it computes only **one** local gradient $\nabla f_i$ at point $x^k$, and the difference between the gradient and memory vector $g_i^k$: $\nabla f_i(x^k)-g_i^k$ due to the use of Error Feedback (EF21) framework:
> > >
> > > 1.  **EF21 Mechanism:** `𝛼-NormEC` leverages EF21 mechanism [Richtárik et al., 2021] to mitigate bias from normalization/clipping. EF21 achieves this by operating on the error-corrected gradient $\nabla f_i(x^k) - g_i^k$, where $g_i^k$ is local error memory. The term communicated to the server *within this framework* is $\mathrm{Norm}_{\alpha}(\nabla f_i(x^k) - g_i^k)$.
> > >
> > > 2.  **DP Requirement:** To ensure privacy, noise must be added to the *communicated* quantity. Thus, we privatize $\mathrm{Norm}_{\alpha}(\nabla f_i(x^k) - g_i^k)$.
> > >
> > > 3.  **Need for $\mathrm{Norm}_{\alpha}$:** The input norm $||\nabla f_i(x^k) - g_i^k||$ is **not** bounded by smoothness alone, as $g_i^k$ accumulates errors and $||\nabla f_i(x^k)||$ can be large (see point 4). Applying smoothed normalization is crucial *within EF21* to guarantee a bounded sensitivity ($||\mathrm{Norm}_{\alpha}(\cdot)|| \leq 1$) for the communicated term, allowing calibrated DP noise addition.
> > >
> > > 4.  **Smoothness vs. Bounded Sensitivity:** We reiterate that smoothness ($||\nabla f(x) - \nabla f(y)|| \leq L ||x-y||$) does **not** imply bounded gradient norm $||\nabla f(x)||$ or bounded $||\nabla f_i(x^k) - g_i^k||$ over an unbounded domain. For instance, considering $y = x^*$ (a stationary point where $\nabla f(x^*) = 0$), smoothness implies $||\nabla f(x)|| = ||\nabla f(x) - \nabla f(x^*)|| \leq L ||x - x^*||$. This bound can be arbitrarily large if $x$ is far from $x^*$. Bounded sensitivity arises from stronger assumptions like Lipschitz continuity of the *function* ($|f(x) - f(y)| \leq l ||x-y||$, implying $||\nabla f(x)|| \leq l$) or a bounded domain. However, assuming Lipschitz continuity restricts the problem class (e.g., excluding quadratic loss) and the constant $l$ is often unknown, making DP noise calibration impractical. Our approach avoids these stronger assumptions by using $\mathrm{Norm}_{\alpha}$ under only standard smoothness.
> > >
> > > 5.  **Alternative Approach:** Privatizing $\nabla f_i(x^k) - \nabla f_i(x^{k-1})$ represents a fundamentally different algorithm that *does not use EF21's bias correction*. Analyzing it would require a separate framework and potentially face different challenges regarding bias accumulation under only smoothness.
> > >
> > > We hope this clarifies our rationale. Thank you again for your constructive engagement.

---

### Official Review · Reviewer_zAsL · 2025-03-15

**Overall Recommendation:** 3

**Summary:**

Clipping the gradients is a common practice in differentially private training with DP SGD and a common technique used to analyze the privacy-utility trade-off of DP-SGD. However, as the authors correctly point out, most theoretical works ignore the effect clipping can have on convergence by assuming bounded gradients and ignoring the effect of clipping altogether. Smoothed normalization is a recent technique that offers a more amenable analysis of DP-SGD without requiring stringent restrictions on the gradients of the clients. This paper shows that smoothed normalization leads to a contractive property on the gradient, making it amenable to the error feedback framework. The paper then analyzes the distributed DP-SGD with smoothed normalization and error-feedback, offering convergence guarantees that illustrate the algorithm privacy-utility trade-off. In doing so, the paper offers an analysis for private smooth non-convex optimization without any bounded gradient assumptions. The paper also offers some experiments comparing their algorithm to other algorithms and testing its sensitivity to hyperparameter tuning.

**Claims And Evidence:**

No. Please see the questions I ask the authors. The claims made about the convergence guarantees in the paper are not accurate and require extreme conditions on the initialization (which seems hard to satisfy without initializing at a point that is already close to stationarity).

**Essential References Not Discussed:**

I can't think of any critical references the authors missed.

**Experimental Designs Or Analyses:**

Yes, the experiments look alright, but I am not convinced that the method is significantly better than Clip21 based on the experiments in Appendix H.4. I believe these experiments should have been included in the paper, and the fact the empirical difference between the two algorithms not high be discussed as a limitation.

**Methods And Evaluation Criteria:**

Yes.

**Other Comments Or Suggestions:**

**1.** The second $L$ should be $L_i$ on line 140 in Assumption 1.

**2.** I am not sure I understand the following comment on lines 184 - 186:

> Thirdly, the condition in (4) is “pathological” in the distributed setting as it restricts the heterogeneity between different clients and can result in vacuous bounds

Why would restricting the data heterogeneity lead to vacuous bounds? Restricting the data heterogeneity should improve the convergence guarantees of at least local update methods. Furthermore, the discrepancy between client gradients could be much smaller than $\phi$, thus avoiding the pessimistic dependence on an enormous $\phi$ in analyzing the "consensus error" between clients.

**3.** Lines 258-266 are written awkwardly and need to be rephrased.

**4.** Why do the authors refer to the initialized memory vectors as $g_i^{-1}$ in lines 288 (column 2) and 301 (column 1) instead of $g_i^0$? This seems to be a typo.

**Other Strengths And Weaknesses:**

The paper is written and identifies a gap in existing literature on private optimization accurately. The proofs are easy to follow and verify as well.

The biggest weakness of the paper is that the final convergence guarantees are too weak, without making very restrictive assumptions about the initialization. I discuss this extensively below in questions for authors. Owing to these limitations, I can not recommend accepting the paper and urge the authors to revisit their analyses to identify any loose inequalities.

**Questions For Authors:**

**Q1.** It is not obvious why it is possible to pick $x^0$ and $g_i^0$ in Corollary 1 to ensure that $\|\nabla f_i(x^0) - g_i^0\|$ is very small. For instance, if the clients' gradients are small ($O(1/\sqrt{K})$) to begin with at $x^0$, then we are done. If this is not the case, one could hope to initialize far away from the optimizers on each machine so that all gradients roughly point in the same direction, but then, if the gradient norms are large, it is not clear if one can find an appropriate $g_i^0$, which could make the difference be much smaller than the gradient norm itself. Could the authors comment on the feasibility of this? To me, it seems like the following inequality in the proof sketch is very loose when we are close to a stationary point, $$\|\nabla f_i(x^k) - g_i^{k+1}\| \leq \max_{i\in[n]}\|\nabla f_i(x^0) - g_i^{0}\|.$$
This constraint in the guarantee almost feels like "cheating."

**Q2.** The criticism of Clip21 in requiring the initial function sub-optimality to tune its step size seems unreasonable. For instance, the choice of $\beta$ after Corollary 1 also depends on the same function sub-optimality. The authors should remove this statement, unless I am missing something? I also disagree with the argument in Appendix E, which essentially relies on making D very small to improve over Clip21. As mentioned above, D can not be made arbitrarily small, at least not without further assumptions about the problem.

**Q3.** The issue raised in Q2 is further exacerbated in Corollary 2. In particular, let's say we want to get to $\epsilon_{err}$ stationarity guarantee. Then, we need to set $R = O(\epsilon_{err})$ (which I argued is already hard). Furthermore, to make the first term small, we need to ensure that $$\frac{L_{max}\alpha (f(x^0) - f^{inf})^2 d^{1/2} \log^{1/2}(1/\delta)}{n^{1/2}\epsilon_{err}^3} \leq \epsilon .$$
With a moderate error rate $\epsilon_{err} = 10^{-2}$, $n=100$ machines and with only $d=100$ dimensions (which is very modest), and $\delta = 10^{-2}$ this implies $\epsilon > 10^6 L_{max}\alpha (f(x^0) - f^{inf})^2$, which is pretty bad. This implies that the privacy-utility trade-off offered by the analyzed algorithm is very poor, and I am not willing to buy that this gives any meaningful privacy guarantee for a reasonable utility. Again, I feel that the presentation and discussion of the results are misleading (hopefully not deliberately). I suspect that, in both analyses, the authors use some very loose inequalities, which causes the final dependence on $R$.

**Relation To Broader Scientific Literature:**

The paper tries to fill the gap in the existing literature by optimizing smooth non-convex functions privately without making any restrictive bounded gradient assumptions. It also addresses this in the novel federated/distributed setting. These analyses require fewer assumptions than existing works but have some significant limitations that I point out below, which require restrictive initialization assumptions.

**Theoretical Claims:**

Yes. The proofs are correct, albeit the main recursion, which bounds $\|\nabla f_{i}(x^k) - g_i^{k+1}\|$ looks pretty loose.

---

> ### Author Rebuttal · Authors · 2025-04-01
>
> Dear reviewer zAsL,
>
> We appreciate your time, effort, and thoughtful feedback.
>
>
> > Q1. It is not obvious why it is possible to pick and  in Corollary 1 to ensure that is very small.
>
> We would like to clarify your misunderstanding. We can initialize $x^0,g_i^0 \in \mathbb{R}^d$ to ensure that $\||   \nabla f_i(x^0)-g_i^0  \||$ is small. For instance, we can choose $x^0$ to be any vector, and $\nabla f_i(x^0)$ does not need to have a small Euclidean norm. Then, we can set $g_i^0=\nabla f_i(x^0) + e$ where $e = (D/\sqrt{K+1}, 0, \dots, 0)$ with any $D>0$ and any total iteration number $K$, and our condition naturally satisfies  $\||\nabla f_i(x^0) - g_i^0 \||= D/\sqrt{K+1}$.
>
> We will include this discussion after Corollary 1 into the revised manuscript.
>
>
> > Could the authors comment on the feasibility of this? To me, it seems like the following inequality in the proof sketch is very loose when we are close to a stationary point,
>
> We kindly disagree that our proof is loose. Existing convergence analysis of Clip21 cannot be applied to prove its convergence in the **private** setting. This motivates us to redesign distributed algorithms using normalization, instead of clipping, to achieve **the first provable utility guarantee in the private setting under smoothness without bounded gradient conditions**. Therefore, our approach for bounding  $\psi^k := \||   \nabla f_i(x^k)-g_i^k  \||$ differs significantly from Clip21, which is heavily based on EF21. We derive the induction proof showing **the monotonicity of $\psi^k$**. This novel condition is **less demanding/restrictive than strong contractivity of $\psi^k$**, i.e. $\psi^{k+1} \leq (1-q) \psi^k, \forall q \in (0, 1]$, which Clip21 and EF21 rely on for variance-reduced Lyapunov-based analysis.
>
>
> > the choice of $\beta$   after Corollary 1 also depends on the same function sub-optimality. The authors should remove this statement, unless I am missing something?
>
> Since $\||   \nabla f_i(x^0)-g_i^0  \||$ can be made small by choosing $x^0,g_i^0\in\mathbb{R}^d$, we have $\|| \nabla f_i(x^k)-g_i^k \||\leq \max_i \||  \nabla f_i(x^0)-g_i^0  \||$ that can be made small. This inequality, and  our step-size rule **do not need the knowledge of the function suboptimality gap**, i.e. $f(x^0)-f(x^\star)$. This is in stark contrast to Clip21, where its step-size rule depends on not only the function suboptimality gap but also $C_1=\max _{i \in[1, n]} \|| \nabla f_i(x^0)\||$. Unlike $\||   \nabla f_i(x^0)-g_i^0  \||$ in $\alpha$-NormEC, $C_1$ in Clip21 cannot be made small easily.
>
>
> > This implies that the privacy-utility trade-off offered by the analyzed algorithm is very poor
>
> Thank you for pointing out our impreciseness. We agree that in the private setting, $R$ cannot be made arbitrarily small, e.g. R goes to zero. We will remove the discussion after Corollary 2, which states that as $R\rightarrow 0$, $\alpha$-NormEC achieves the utility bound of $\mathcal{O}\left( \Delta \sqrt[4]{ \frac{d\log(1/\delta)}{n\epsilon^2} } \right).$
>
>
>
> > Yes, the experiments look alright, but I am not convinced that the method is significantly better than Clip21 based on the experiments in Appendix H.4. I believe these experiments should have been included in the paper, and the fact the empirical difference between the two algorithms not high be discussed as a limitation.
>
> We kindly disagree as our main focus is on private training. First, our method has proved convergence guarantees unlike Clip21 in the private training. Second, we do not claim that our method outperforms Clip21  in the non-private setting; however, Table 3 shows that $\alpha$-NormEC achieves slightly higher accuracy than Clip21 for most values of $\beta \in \{0.01, 0.1, 1\}$.
>
> We performed additional comparisons in  the private training, where DP-$\alpha$-NormEC significantly outperforms DP extension of Clip21 (which does not have convergence guarantees in the private setting).  Kindly see [the attached plot link](https://postimg.cc/bDhgkq7R).
>
>
> > Why would restricting the data heterogeneity lead to vacuous bounds?
>
> Bounded heterogeneity $\delta:=\|| \nabla f_i(x) - \nabla f_j(x)\||$ is an unrealistic condition, as heterogeneity can be arbitrarily large in practice. As $\delta$ grows, any convergence bounds that depend on $\delta$ can be very loose (**vacuous bounds**). This implies that the corresponding algorithms do not converge. Furthermore,the bounded gradient assumption implies bounded heterogeneity. We kindly refer to Khaled et al. (2020) for the discussions in more detail.
>
>
> Furthermore, thank you for spotting the typos. We will fix them in the revision.
>
>
> We hope these clarifications have sufficiently addressed your concerns, providing a clearer understanding of our study's contributions and implications. We are eager to engage in further discussion to resolve any remaining concerns.
> Please consider the score accordingly.
>
>
> Best regards,
>
> Authors

---

### Decision · Program_Chairs · 2025-05-01

**Decision:**

Reject

**Comment:**

This paper introduces α-NormEC, addressing the convergence limitations of clipping-based DP-SGD by combining smoothed normalization and error feedback, and removing restrictive bounded gradient assumptions. Reviewers recognized the theoretical rigor and practical novelty but noted limitations regarding theoretical strength under realistic privacy settings, experimental generalizability, and comparison to other adaptive clipping methods in the literature.
As the authors acknowledged, the private version of this algorithm, which the the stated main motivation, the bounds depend on the parameter "R" and it is not clear to how one can set the initial g's privately to make this small (or go to zero as n goes to infty). This seems to contradict the assertion about convergent bounds under DP in the abstract. The authors are encouraged to explain better where the initial g_i's come from (both in theory and in their experiments), and address other feedback from the reviewers. In the current form, the paper does not meet the bar for this conference.